# ThunderAgent: A Fast, Simple, and Program-Aware Agentic Inference System

Hao Kang [* 1] Ziyang Li [* 2] Weili Xu [* 3] Xinyu Yang [* 4] Yinfang Chen [3] Junxiong Wang [5] Beidi Chen [4] Tushar Krishna [1] Chenfeng Xu [5] Simran Arora [5]

## Abstract

Large language models (LLMs) are now used to power complex multi-turn agentic workflows. Existing systems run agentic inference by loosely assembling isolated components: an LLM inference engine (e.g., vLLM) and a tool orchestrator (e.g., Kubernetes). Although agentic workflows involve multiple LLM and tool requests, these systems schedule and allocate resources separately on a per-request basis, without end-to-end knowledge of the workflow. This leads to sub-optimal management of KV cache and tool execution environments. To address the challenges, we propose THUNDERAGENT, a fast, simple, and program-aware agentic inference system. We first abstract agentic workflows as *LLM Programs*, enabling a unified view of heterogeneous resources, including KV caches, system states, and external tool assets such as disk memory and network ports. Built upon this abstraction, THUNDERAGENT introduces a program-aware scheduler and a tool resource manager designed to maximize KV cache hit rates, mitigate memory imbalances, and enable asynchronous environment preparation. Evaluations across coding, routing, and scientific discovery agents demonstrate that THUNDERAGENT achieves **1.5-3.6×** throughput improvements in serving, **1.8-3.9×** in RL rollout, and up to **4.2×** disk memory savings compared to state-of-the-art inference systems. To facilitate reproducibility and support future development, we open-source the system implementations of THUNDERAGENT at: https://github.com/ThunderAgent-org/ThunderAgent.

---

*Equal contribution   [1]Georgia Institute of Technology [2]Individual Researcher [3]University of Illinois Urbana-Champaign [4]Carnegie Mellon University [5]Together AI. Correspondence to: Hao Kang <hkang342@gatech.edu>.

*Proceedings of the 43rd International Conference on Machine Learning*, Seoul, South Korea. PMLR 306, 2026. Copyright 2026 by the author(s).

## 1. Introduction

Recent advances in language models have expanded their use beyond basic chatbots to complex agents (Yang et al., 2025; Team et al., 2025). These agents address real-world problems in domains such as coding (Jimenez et al., 2024; Jain et al., 2024) and computer-use (Xie et al., 2024; Bonatti et al., 2024) by interleaving long reasoning with external tool calls (e.g., compilers, retrievers), often operating as autonomous systems that execute multi-step workflows without real-time human intervention. However, the throughput of modern inference systems degrades as the number of agentic requests being processed increases (Figure 1a). Meanwhile, rollout accounts for over **70%** of the total wall-clock time in reinforcement learning (RL) (Sheng et al., 2025; Fu et al., 2025).

As agentic workflows become increasingly autonomous at scale, overall system efficiency is governed by sustained throughput rather than tail latency, whereas human-in-the-loop applications are often dominated by user response times. Therefore, higher throughput directly reduces serving cost by amortizing hardware over more completed workflows. Moreover, in asynchronous RL, higher rollout throughput mitigates policy lag between the parameters used for data collection and those being updated. This allows the model to learn from data with reduced staleness, improving both convergence speed and final policy quality (Fu et al., 2025; Sheng et al., 2025; Shenfeld et al., 2025; Zheng et al., 2025).

However, current agentic inference systems provide sub-optimal throughput because they are loosely combined from isolated components: an off-the-shelf model inference engine(e.g., vLLM (Kwon et al., 2023) or SGLang (Zheng et al., 2024)) coupled with a general-purpose tool orchestrator (e.g., Kubernetes). While agentic workflows involve multiple turns of model and tool requests, these components schedule and allocate resources separately on a *per-request basis*, without end-to-end knowledge of the entire workflow. This design gives rise to three key challenges:

1. **KV cache thrashing.** Under memory pressure at high concurrency, request-aware systems reactively evict the KV cache of programs in their tool-execution intervals to make room for decode requests of other concurrent programs, without foresight into future reuse in the agent

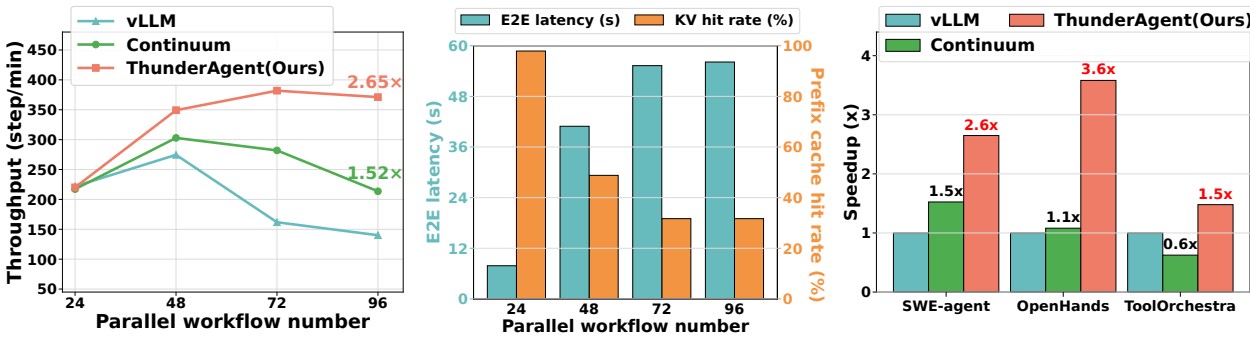

*(a)* Throughput degradation      *(b)* KV cache thrashing      *(c)* Speedup from THUNDERAGENT

*Figure 1.* **Performance comparison of THUNDERAGENT against prior agent inference systems as the parallel workflow number (i.e., batch size) increases.** We evaluate the GLM-4.6 MoE model serving SWE-Agent on SWE-Bench Lite (Figures a and b) and SWE-Agent, OpenHands, and ToolOrchestra (Figure c) on an 8×H100 GPU cluster. Results show that: (a) Current inference systems fail to maintain high throughput at large batch sizes. (b) Throughput degradation is primarily caused by low KV cache hit rates, which increase end-to-end request latency. (c) THUNDERAGENT achieves high throughput compared to prior inference systems by reducing KV-cache thrashing and managing the lifecycle of tool execution resources.

workflow. Thus when the tool call completes, the system needs to rerun prefill to recover its whole interaction history. The re-prefill cost increases the average end-to-end latency of agent workflows by up to **7.14×** (see Figure 1b) and decreases throughput.

2. **Cross-node memory imbalance.** The request-aware engines suffer from imbalanced utilization in multi-node inference setups. Existing engines pin all requests from the same agentic workflow to a fixed node to maximize the KV cache hit rate. However, as context lengths scale rapidly and unpredictably in agent workflows, some nodes reach capacity while others remain underutilized under this routing policy.

3. **Tool lifecycle obliviousness.** The request-aware orchestrators struggle to decide when to release and prepare resources and environments required for tool execution. Thus, unused sandboxes and API servers continue to occupy critical disk space and network ports, leading to cumulative resource exhaustion and system failures. Meanwhile, agentic workflows have to wait for extremely long setup time before reasoning.

This work introduces THUNDERAGENT, an agentic inference system that adopts an end-to-end view of agentic workflows to enable high-throughput agentic serving and RL rollout. Our specific contributions are:

1. **Program abstraction:** We abstract agent workflow as *agentic programs*. An agentic program is a first-class scheduling unit that persists across multiple model invocations and tool executions, exposing semantic state to the runtime. A program tracks metadata for the workflow's identifier, execution phase (i.e., reasoning or acting), scheduling status, total tokens, and tool resources. This abstraction decouples scheduling from execution backends (e.g., vLLM/SGLang/TensorRT-LLM), enabling seamless integration of new workflows.

2. **Program-aware scheduler:** Based on the program abstraction, we cast agentic inference scheduling as a constrained optimization problem to minimize the recomputation and caching overheads, and maximize prefilling and decoding throughput, subject to GPU memory capacity. We introduce two key mechanisms:

   (a) **State-aware pausing:** If the execution backend experiences memory pressure, we selectively pause workflows that are currently in the acting state with tool call. This design helps preserve memory for programs that are in the reasoning state and eliminate arbitrary, sub-optimal KV cache eviction.

   (b) **Dynamic migration:** We migrate agent programs across data parallel (DP) GPU nodes to mitigate memory imbalance. We accomplish this by enabling all DP nodes share a global program-aware waiting queue, rather than enforcing that requests from a program are always sent to the same node.

3. **Program-aware tool resource management:** In long-horizon agentic workloads, tool environments are persistent resources whose mismanagement directly limits sustained throughput. By tracking execution dependencies, THUNDERAGENT overlaps I/O-intensive environment initialization with LLM reasoning. For completed programs, we implement a lifecycle-aware garbage collector that leverages program termination signals to reclaim tool resources such as Docker sandboxes and network ports. Consequently, this prevents accumulated resource leakage and ensures sustained high-throughput agentic inference in THUNDERAGENT.

The above contributions cannot be achieved within request-aware inference engines. Without an explicit representation of program states and workflow dependencies, request-aware schedulers cannot distinguish temporary tool waits from termination or coordinate GPU memory with program-level resource scheduling.

We evaluate THUNDERAGENT across diverse agentic workloads. For **serving**, we evaluate the ToolOrchestra (Su et al., 2025) as routing agent on HLE-Bench (Phan et al., 2025), SWE-Agent (Yang et al., 2024) and OpenHands (Wang et al., 2025b) as coding agent on SWE-bench (Jimenez et al., 2024), and OpenHands as scientific discovery agent on ScienceAgentBench (Chen et al., 2024), achieving **1.48–3.58×** throughput improvements as illustrated in Figure 1c. For **RL rollouts**, we further test the coding agents on distributed GPU nodes, achieving **1.79–3.92×** improvements compared with prior SOTA systems.

Beyond these results, we evaluate THUNDERAGENT under common large-scale deployment settings. As detailed in the appendix, THUNDERAGENT's throughput scales near-linearly with multiple worker replicas on up to 64 H100 GPUs (Section A.3), and remains effective when combined with KV-cache offloading (Section A.4). THUNDERAGENT has been adopted by open-source frameworks such as SkyRL and NVIDIA Dynamo (Section A.6).

## 2. Background

In this section, we provide background on the properties and existing approaches to support agentic inference.

### 2.1. System Properties of Current Agentic Workflows

Current agentic workflows alternate between reasoning and acting during generation. Formally, at each step $t$, the agent receives an observation $o_t \in \mathcal{O}$ and produces an emission $e_t = (\ell_t, a_t) \in \mathcal{L} \times \mathcal{A}$, where $\ell_t$ denotes a thought and $a_t$ represents an action. We define the cumulative context at step $t$ as $c_t = (o_1, e_1, \ldots, o_t)$, which captures the interaction history of agentic workflows. Conditioned on $c_t$, $e_t$ is sampled from a policy $\pi(e_t|c_t)$.

This workflow keeps two persistent states: *(i) GPU Memory*, where the KV cache of $c_t$ serves as the workflow's memory. As the trace grows incrementally, $c_{t+1}$ extends $c_t$ as a prefix, enabling theoretical near-complete KV cache reuse rates across steps. *(ii) Tool Environment*, where external resources (e.g., sandboxes) initialized at $t = 1$ must remain consistent and accessible throughout the execution.

These stateful dependencies necessitate a *program-level* view of agentic inference trajectories, thereby enabling to system to coordinate heterogeneous resources and manage state across long-running workflows. However, existing inference systems treat each thought $l_t$ and action $a_t$ as an independent, stateless request.

### 2.2. Existing Agentic Inference Systems

Prior work focuses on optimizing the individual components in agentic inference, including the LLM inference engine or tool orchestrator (Section A.1, Section A.2), but there are very few works that provide end-to-end optimization for agentic workflows across GPU, CPU, and remote resources.

Autellix models multi-turn agentic workflows as **GPU-only programs** and tracks the accumulated GPU execution time in a central process table (Luo et al., 2025). However, it ignores workflow locality, allowing concurrent workflows to aggressively evict other's KV cache, triggering **KV cache thrashing** under heavy workloads.

Continuum is another recent serving system designed for multi-turn agentic workflows (Li et al., 2025). It employs a time-to-live (TTL) mechanism to pin KV caches in HBM, thereby mitigating context thrashing during tool execution. However, it fails to solve the KV cache eviction problem. The first reason is that most tools take an **unpredictable** amount of time (e.g. remote model APIs in ToolOrchestra (Su et al., 2025), compilers in code agents, and web applications for computer use agents (Zhou et al., 2024)). Such unpredictable tools trigger severe thrashing as well as stranded KV cache memory in Continuum due to incorrect TTL estimates. Moreover, once the decoding memory of the running workflow surpasses the GPU limit, the system preempts and evicts pinned KV cache as well. This leads to unavoidable thrashing and corresponding throughput degradation shown in Figure 1a.

These limitations underscore the need for a simple and fast system for agentic inference. We envision such a system as a program-aware scheduling layer for emerging agentic inference systems (e.g., Zhang et al. (2026)).

## 3. Challenges in Existing Agentic Inference Systems

This section profiles vLLM combined with Kubernetes as a representative baseline for multi-turn agentic inference, and synthesize its key inefficiencies. Notably, the identified limitations cannot be solved by replacing the inference engine, but rather require new program-aware abstractions. By default, we use GLM 4.6 model for OpenHands RL rollout on two 8×H100 GPU nodes.

### 3.1. KV Cache Thrashing

Agentic workflows exhibit a high theoretical KV cache reuse rate during their execution. However, in existing LLM serving systems, each step is served as an independent and stateless request. Under high concurrency, this request-level scheduling causes KV cache to be frequently evicted during tool execution to accommodate newly arriving requests, resulting in repeated eviction and reprefill, which we refer to as **KV cache thrashing**.

As shown in Figure 1b, this thrashing intensifies as the number of parallel workflows increases. The resulting degradation in cache hit rates triggers frequent and costly re-prefill, where the prefix must be recomputed upon tool completion. This redundancy significantly increases the end-to-end latency of each request by up to **7.14×** compared to a non-thrashing setting, leading to severe throughput degradation.

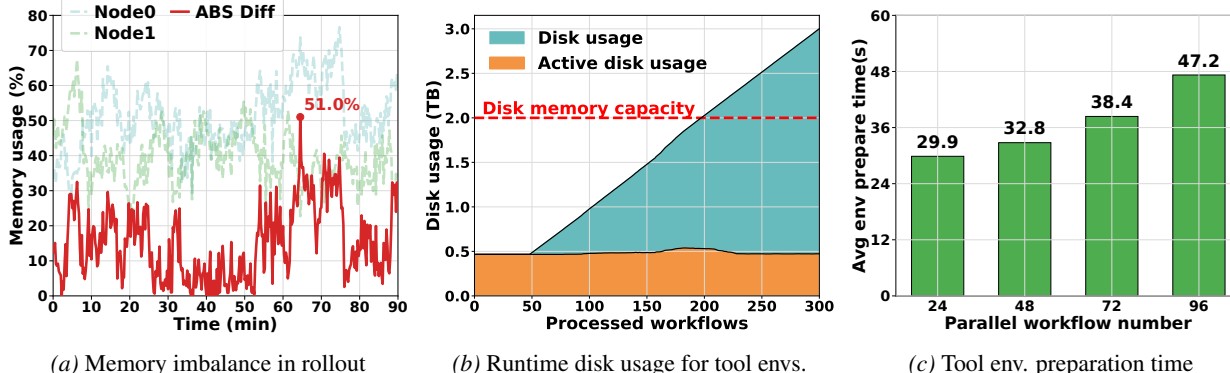

*(a)* Memory imbalance in rollout     *(b)* Runtime disk usage for tool envs.     *(c)* Tool env. preparation time

*Figure 2.* **Demonstrations of the memory imbalance and tool resource management problems for current agentic inference systems.** We evaluate vLLM + Kubernetes on OpenHands RL rollout using the GLM 4.6 model on SWEBench-Lite with two $8\times$H100 GPU Nodes. The observations show: (a) Max memory imbalance can achieve 51% on 90 min rollout tests when applying vLLM KV-aware router. (b) Failure to garbage collect tool execution environments gradually causes resource usage to exceed system capacity. (c) Average tool execution environment preparation time grows fast as parallel workflow number increases.

### 3.2. Cross-Node Memory Imbalance

Current policies for routing requests across worker replicas are also sub-optimal. Existing multi-turn schedulers (Zheng et al., 2024; vLLM Team, 2025) greedily assign requests to the target with the highest KV-cache locality in order to maximize cache reuse. However, this policy ignores the fact that the memory load can become imbalanced across nodes. For instance, the KV-aware router in vLLM (vLLM Team, 2025) sends all requests from the same agentic workflow to the same node. Since different workflows can exhibit highly heterogeneous KV footprints and execution lifetimes, this policy often results in severe memory imbalance across nodes, with some nodes are overloaded while others remain lightly utilized. Similarly, the cache-aware router in SGLang routes via a router-side radix tree that approximates worker KV state. Under high agentic concurrency, worker-side evictions from KV thrashing leave this tree stale, yielding both poor cache reuse and cross-node imbalance.

As shown in Figure 2a, during a 90 minute snapshot of agentic RL rollout, the memory usage between two DP nodes diverges by **more than 20% for over 37 minutes, reaching a peak imbalance of 51%**.

### 3.3. Tool Lifecycle Obliviousness

Current agentic inference systems do not synchronize the external tool orchestrator's lifecycle with the LLM inference engine, resulting in silent resource wastage and latency overhead on the tool orchestrator side.

**Resource leakage and unused sandboxes.** Figure 2b showcases that the total disk space consumption increases linearly with the number of processed workflows, eventually exceeding system capacity. This is because unused resources (e.g., Docker images of finished workloads) are not reclaimed when workflows complete. This inefficient garbage collection leads to fatal system instabilities for long-running agentic inference workloads.

**Costly environment preparation.** We observed that most agentic workloads need to prepare environments before initiating the multi-turn trajectory. For example, coding agents need to pull dockers, install related packages and build repositories. Furthermore, this preparation time is costly and increases with the parallel workload number, as shown in Figure 2c. If the LLM inference engine needs to wait until the environments are fully prepared, this overhead will extend the end-to-end latency of the inference system.

## 4. THUNDERAGENT: A Program-Aware Agentic Inference System

With all findings in Section 3, we present THUNDERAGENT, a program-aware system for high throughput agentic inference. We model the Agentic Program in Section 4.1, which serves as our primary abstraction for scheduling. Section 4.2 formalizes a cost model to guide our system design. Built upon these foundations, we detail our KV cache scheduling policy in Section 4.3 and tool resource management strategy in Section 4.4.

### 4.1. Program Abstraction

The **Agentic Program** serves as a fundamental abstraction that encapsulates both the logical execution flow and the system-level dependencies of agentic workflows. Formally, we define an agentic program $P$ as a tuple:

$$P = \langle ID, c, \mathcal{T}, \mathcal{L}, \tau, s \rangle, \tag{1}$$

where $ID$ represents the unique global identifier. $c$ denotes the number of tokens in the context, corresponding to the KV cache memory footprint during active execution. $\mathcal{T}$ tracks the set of tool environments used by the program, enabling garbage collection when no program requires them further. $\mathcal{L}$, $\tau$, and $s$ denote the node placement, execution phase, and scheduling status, respectively, facilitating program-level KV cache thrashing reduction and cross-node transferring. A metadata example

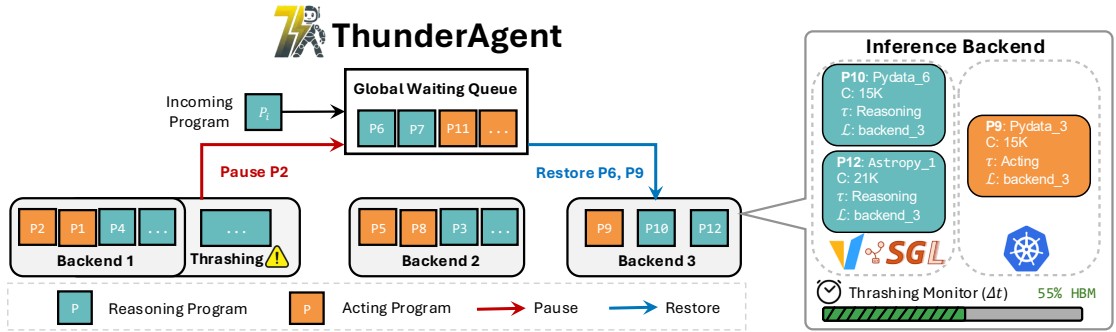

*Figure 3.* **An Overview of THUNDERAGENT.** We show the transition between scheduling states and memory management. THUNDERA-GENT queries the state of each data parallel backend periodically every $\Delta t$ time. Here, Backend #1 triggers thrashing, while Backend #3 is underutilized. The global waiting queue shared by all Backends then pauses and collects acting Program #2 back to the queue while releasing reasoning Program #6 and #9, to stop the KV-cache thrashing in Backend #1 and reduce memory imbalance of Backend #3.

is demonstrated on the right side of Figure 3.

THUNDERAGENT directly wraps existing LLM engines and tool orchestrators by interfacing with OpenAI-style endpoints. Program IDs allow the system to distinguish requests from different agentic workflows. We elaborate on the simplicity of integrating THUNDERAGENT with existing inference services in Appendix B.

*Table 1.* **Summary of Notations for agentic programs.** Each program instance is characterized by its identity, execution phase, tool environments, resource footprint, and scheduling state.

| Notation | Description |
|---|---|
| $P$ | Agentic program instance |
| $ID$ | Unique global identifier for the program |
| $c$ | Number of tokens in the context |
| $\mathcal{T}$ | Set of tool environments required by the program |
| $\mathcal{L}$ | Backend placement for cache affinity |
| $\tau$ | Execution phase: Reasoning (**R**), Acting (**A**) |
| $s$ | Scheduling status: {Active, Paused, Terminated} |

### 4.2. Cost Model

During multi-turn agentic inference, only the resources used for active prefilling and decoding contribute to the system's effective throughput, while re-computation, used capacity, and idle caching constitute resource waste. We encompass this in a cost model for GPU resource consumption, which isolates effective costs from non-productive usage. We adopt the Space-Time Product (STP) (Belady, 1966) as our primary metric, defined as the integral of the memory footprint over processing time. The STP cost during a process phase is formalized as:

$$\text{Cost}_{\text{x}} = \int_0^{t_x} M_x(t)\, dt, \tag{2}$$

where $t_x$ is the duration of process $x$ (e.g., prefill). Since memory usage $M_x(t)$ can be directly quantified by the KV cache token count used in LLMs, we define our cost model as the integral of token count over time.

The total cost of agentic inference comprises five distinct components: decoding, prefilling, recomputation, unused

capacity, and idle caching. We explicitly distinguish incremental prefilling for tool execution results from recomputation over historical interactions, with the latter leading to significantly higher cost due to re-computing evicted KV cache over the full context. Formally, this yields the following cost decomposition:

$$\text{Cost}_{\text{total}} \approx \text{Cost}_{\text{decode}} + \text{Cost}_{\text{prefill}} + \text{Cost}_{\text{recompute}} + \text{Cost}_{\text{unused}} + \text{Cost}_{\text{caching}} \tag{3}$$

In this decomposition, $\text{Cost}_{\text{decode}}$ and $\text{Cost}_{\text{prefill}}$ represents the effective work that contributes to inference throughput. The remaining terms are wasted system overheads: $\text{Cost}_{\text{recompute}}$ stems from KV cache thrashing (Section 3.1); $\text{Cost}_{\text{unused}}$ reflects memory imbalance across data parallel (DP) inference backend replicas (Section 3.2); and $\text{Cost}_{\text{caching}}$ accumulates while holding memory during external tool execution (Section 3.3).

### 4.3. Scheduling Policy

Based on the cost model above, the optimization target of our scheduling policy is to minimize the non-productive overhead components: $\text{Cost}_{\text{recompute}}$, $\text{Cost}_{\text{unused}}$, and $\text{Cost}_{\text{caching}}$, thereby maximizing throughput.

#### 4.3.1. REDUCING RECOMPUTATION AND CACHING COSTS VIA PROGRAM-AWARE WAITING QUEUE

As identified in Section 3.1 and Figure 1b, KV cache thrashing serves as the primary bottleneck for throughput degradation. To address this limitation, the system must minimize $\text{Cost}_{\text{recompute}}$ by explicitly controlling the number of active programs. THUNDERAGENT achieves this by introducing a program-aware waiting queue. Our system utilizes this queue to schedule program execution, determining which program should be executed in GPU versus which should be swapped out based on their token length $c$ and execution phase $\tau$. Here, we formalize the scheduler behavior using two primitive operations: **Restore** and **Pause**, as follows.

- **Restore.** This operation admits a program into active execution. Given a program $P = \langle ID, c, \mathcal{T}, \mathcal{L}, \tau, s \rangle$ with $s = \text{Paused}$ and $\mathcal{L} = \varnothing$, Restore$(P)$ assigns $P$ to a

backend $\mathcal{L}'$ with available capacity and updates

$$P \leftarrow \langle ID, c, \mathcal{T}, \mathcal{L}', \tau, \text{Active} \rangle, \tag{4}$$

- **Pause.** This operation removes a program from active execution. Given a program $P = \langle ID, c, \mathcal{T}, \mathcal{L}, \tau, s \rangle$ with $s = \text{Active}$, $\text{Pause}(P)$ unbinds $P$ from its backend, releases its KV cache for preemption, and updates

$$P \leftarrow \langle ID, c, \mathcal{T}, \varnothing, \tau, \text{Paused} \rangle. \tag{5}$$

Building on these two operations, we next introduce our scheduling policy to minimize KV cache thrashing.

**Periodic thrashing detection.** The program abstraction in Section 4.1 provides us with the KV cache size of acting programs. Notably, this is unavailable in request-level systems (as in Section 3). We define the thrashing condition for a DP backend $\mathcal{L}$ as the state where program memory demand exceeds total capacity:

$$C_{\text{total}} < \sum_{p \in \mathcal{L}} c_p \tag{6}$$

where $C_{\text{total}}$ denotes the fixed token capacity of the KV cache pool for backend $\mathcal{L}$. During decoding, the context length $c_p$ of agentic workflows grows rapidly, which can trigger memory thrashing *mid-execution* even without of new arrivals. Unlike baseline schedulers (e.g., Continuum) that only perform checks on whether to admit a workflow upon its arrival, we implement a **periodic monitor** that evaluates the memory usage at fixed intervals $\Delta t$, allowing preemptive detection and mitigation of memory pressure caused by context growth.

When KV cache thrashing is imminent, THUNDERAGENT invokes **Pause** operation to suspend active programs and free memory size $\Delta C = \sum_{p \in \mathcal{L}} c_p - \lambda_{\max} \cdot C_{\text{total}}$ until the total memory usage falls below the limit $\lambda_{\max} \cdot C_{\text{total}}$. Conversely, when the backend has available space, meaning $\sum_{p \in \mathcal{L}} c_p < \lambda_{\min} \cdot C_{\text{total}}$, THUNDERAGENT restores paused programs from the waiting queue via **Restore**, ensuring that the restored program keeps the total memory below $\lambda_{\max} \cdot C_{\text{total}}$. Here, $\lambda_{\max}$ and $\lambda_{\min}$ denote the high- and low-watermarks of memory usage, respectively, together forming a hysteresis window that stabilizes our scheduling. In practice, we set both value to be 1, as the shared prompt across programs implicitly reserves sufficient memory buffer.

With this program-level periodic capacity check, THUNDER-AGENT can guarantee that there will be no KV cache thrashing by reserving memory for active programs during the acting phase. However, the tradeoff is that when programs engage in long-running tool execution, the GPU memory occupied by acting programs is idle. To balance the cost of caching against recomputation, we incorporate a time-decay mechanism into the thrashing check that progressively discounts the effective weight of acting programs' tokens. This allows the scheduler to evict long-idling caching when memory pressure rises, rather than holding them indefinitely:

$$C_{\text{total}} < \sum_{p \in \mathcal{L}, \tau = \mathbf{R}} c_p + \sum_{q \in \mathcal{L}, \tau = \mathbf{A}} c_q \times f(t_q) \tag{7}$$

Specifically, $t_q$ is the tool execution time of program $q$ in the current step. $f(t)$ is a time-decay function designed to balance $\text{Cost}_{\text{caching}}$ and $\text{Cost}_{\text{recompute}}$. Note that this decayed check governs both **Pause** and **Restore**: shrinking an idle program's weight admits more reasoning programs, whose decoding then raises pressure and triggers the eviction of its KV cache. By dynamically lowering the effective memory priority of acting programs over time, $f(t)$ encourages the scheduler to evict caches that remain idle. In Section F.1, we prove that when tool execution latencies satisfy the memoryless property (i.e., the remaining execution time is independent of the elapsed duration), exponential decay is the only admissible form of $f(t)$.

**Minimizing $\text{Cost}_{\text{recompute}}$ via Shortest-First Eviction.** With the eviction and restoration conditions above, the remaining question in handling thrashing is to determine which subset of active programs to pause such that the recomputation cost is minimized. In this paragraph, we demonstrate that evicting programs with the smallest KV cache size yields the optimal solution, with a detailed proof provided in Section F.2.

**Lemma 4.1** (Quadratic Recomputation Cost). *Given a program $P_i$ with context length $c_i$, the recomputation cost incurred by reprefilling its KV cache scales quadratically with $c_i$, i.e.,*

$$Cost_{recompute} = \int_0^{t_{\text{recompute}}} c_i(t)\, dt \propto c_i^2. \tag{8}$$

**Definition 4.2** (Eviction Optimization Problem). Based on Lemma 4.1, given a required memory release $\Delta C$, the scheduler aims to select a subset $S$ of programs to evict such that the released capacity satisfies the constraint while minimizing the total recomputation cost. This optimization problem is formulated as follows:

$$\min_S \sum_{i \in S} c_i^2 \quad \text{s.t.} \quad \sum_{i \in S} c_i \geq \Delta C. \tag{9}$$

The objective is strictly minimized by selecting smaller $c_i$. Thus, THUNDERAGENT's strategy is to greedily pause and evict programs with the **shortest context lengths**. We defer the formal proof to Appendix F.3. Based on these analyses, we employ the following scores for restoring and pausing a program in our scheduler:

$$S_{\text{restore}}(P) = \frac{1}{c_P} + \mathbb{I}(\tau = \mathbf{R}) \tag{10}$$

$$S_{\text{pause}}(P) = \frac{1}{c_P} + \mathbb{I}(\tau = \mathbf{A}) \qquad (11)$$

where the indicator function $\mathbb{I}(\cdot)$ enforces strict prioritization of the program's execution state ($\tau$) over context length. Both mechanisms follow the **shortest-first** policy to minimize recomputation cost. However, the state indicator $\mathbb{I}$ ensures that the scheduler prioritizes pausing Acting programs, thereby minimizing $\text{Cost}_{\text{caching}}$ by reclaiming cached memory, while prioritizing restore Reasoning programs to maximize $\text{Cost}_{\text{decode}} + \text{Cost}_{\text{prefill}}$.

#### 4.3.2. REDUCING MEMORY IMBALANCE VIA GLOBAL PROGRAM-AWARE WAITING QUEUE

Section 3.1 and Figure 2a highlight that memory imbalance across nodes introduces significant $\text{Cost}_{\text{unused}}$, leading to unnecessary program pausing despite sufficient memory capacity from other nodes. To this end, THUNDERAGENT unify waiting queues of all backend replicas into a **global program-aware waiting queue**.

The key motivation of this design is that $\text{Cost}_{\text{unused}}$ arises only when paused programs remain in the waiting queue while some replicas have idle memory. Moreover, once a program is paused, its KV cache is assumed to be evicted, making its recomputation cost node-agnostic. This allows us to improve cross-node memory balance without sacrificing KV cache locality. The restore policy aligns with load balancing rather than strict KV-aware routing, enabling paused programs to be resumed on the worker replica with available memory capacity. As a result, the global queue bounds the unused cost such that $C_{\text{unused}} < c_{\text{min}} \cdot \Delta t$[1] for every node in the period of $\Delta t$, where $c_{\text{min}}$ represents the minimum token length among paused programs. An overview of the scheduling policy and the global waiting queue in THUNDERAGENT is presented in Figure 3.

### 4.4. Tool Resource Management

Next, THUNDERAGENT mitigates the resource leakage and environment setup overheads detailed in Section 3.3.

**Hook-based garbage collection.** We implement lifecycle hooks that strictly couple the persistence of tool resources with the agentic program's scheduling status $s$. When a program is *Terminated*, the collector triggers an immediate teardown sequence, systematically reclaiming sandboxes, network sockets, and compute slots. The active disk usage in Figure 2b shows that our resource management policy effectively prevents the accumulation of excessive resources and maintains near-constant disk memory consumption over time.

**Asynchronous environment preparation.** The latency involved in initializing a tool execution environment (e.g., starting a Docker container and installing dependencies) can

---

[1]Since $\Delta t$ is much smaller than a program's lifetime, we ignore the impact of terminated programs within a single interval.

be a bottleneck. To address this, THUNDERAGENT monitors the global waiting queue; when a high-priority program (high $S_{\text{restore}}$) approaches the restore threshold, the system asynchronously restores its execution environment before the GPU memory is allocated. This technique effectively **hides the initialization overhead**, significantly reducing end-to-end latency for tool-call heavy workloads like coding agents and science agents, as demonstrated in Figure 2c.

## 5. Experiments

In this section, we evaluate THUNDERAGENT on diverse agentic workflows, including coding, routing, and scientific research agents, and RL rollout across multiple hardware configurations ranging from RTX5090 to H100 clusters. Furthermore, we conduct extensive ablation studies to breakdown the end-to-end system runtime and to describe the system's sensitivity to the scheduler's hyperparameters, $\Delta t$ and $f(t)$, in Section 5.4.

### 5.1. Experimental setup

**Benchmarks and workflows.** We evaluate THUNDERAGENT against diverse benchmarks and workloads:

1. **Coding agent serving.** We deploy **OpenHands** and **mini-SWEAgent** on the SWEBench-Lite (Jimenez et al., 2024). OpenHands represents a **heavy-initialization workflow** with an average disk footprint exceeding 10GB per sandbox, while mini-SWEAgent is a **lightweight workflow** with a minimal footprint (2GB per sandbox).

2. **General agent serving.** We apply ToolOrchestra on HLE (Phan et al., 2025) and OpenHands on ScienceAgentBench (Chen et al., 2024). These workloads involve variable latencies driven by external API calls and complex scientific simulations.

3. **RL rollout.** We apply the same models, workflows, and samples for RL rollout on two $8\times$ H100 nodes.

**Models and deployments.** We employ GLM-4.6 (355B) (Team et al., 2025) and Qwen-3 (235B) (Yang et al., 2025) using both OpenHands (Wang et al., 2025b) and mini-SWEAgent (Yang et al., 2024) frameworks. Models are quantized to FP8 with Tensor Parallelism (TP8) on $8\times$H100 nodes. For ToolOrchestra (Su et al., 2025), we use Qwen3-8B with FP16 precision hosted on one RTX 5090. We deploy the LLM inference engine and Docker at different clusters. The LLM inference engine runs on GPU clusters hosting the models, while agent Docker environments are offloaded to a dedicated CPU cluster.

**THUNDERAGENT configuration.** We configure THUNDERAGENT with hyperparameters $\Delta t = 5$ and priority decay $f(t) = 2^{-t}$, defined in Section 4.3. vLLM is employed as our LLM inference engine. We use steps per minute as our throughput metric, where one step includes a reasoning and acting period of the workflow.

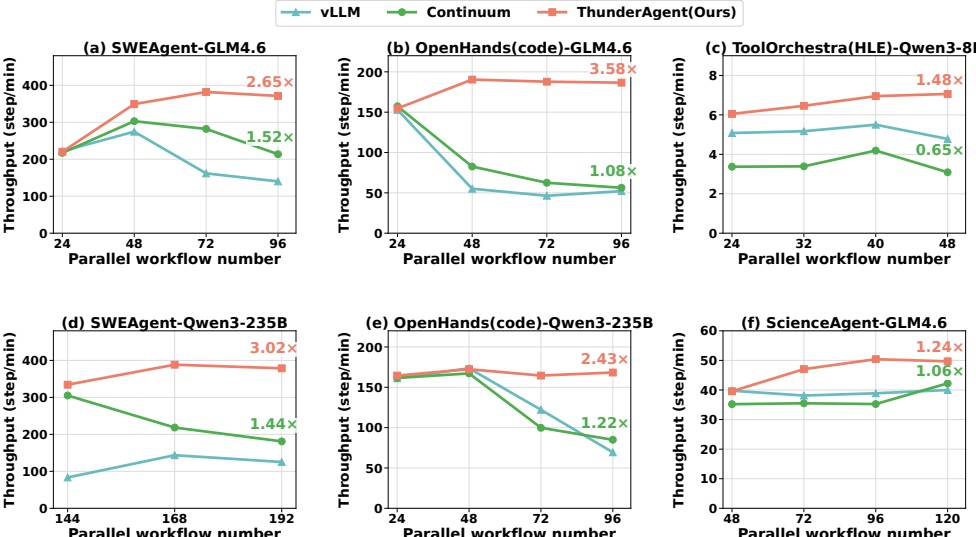

*Figure 4.* **Serving Evaluation Results.** THUNDERAGENT outperforms vLLM and Continuum across three models, four agentic workflows, and three datasets. For workflows with predictable tool call times (a, b, d, e), THUNDERAGENT outperforms vLLM and Continuum up to 2.43-3.56×. For workflows exhibit stochastic tool execution time (c, f), THUNDERAGENT still achieves the best throughput performance.

**Baseline techniques.** We compare against state-of-the-art systems with different scheduling paradigms:

- **vLLM (Inference):** A widely adopted, request-aware LLM inference engine that serves as a stateless baseline for inference performance, without incorporating any agent- or program-specific awareness.

- **Continuum (Inference):** The current SOTA system for multi-turn agentic workflows. It mitigates KV cache thrashing by predicting tool execution durations and pinning KV cache to HBM correspondingly.

- **vLLM + SGLang Gateway (Distributed Rollout):** The leading solution for large-scale distributed RL rollout. SGLang Gateway optimizes distributed inference by enhancing cross-node memory balancing and KV cache hit rates, making this combination a strong baseline for the distributed RL rollout setting.

### 5.2. Serving Evaluation Results

**High throughput under high concurrency.** Figure 4 showcase that THUNDERAGENT demonstrates superior throughput at high concurrency levels (e.g., 96 parallel programs), achieving a **1.48–3.58×** speedup over vLLM and **1.17–3.31×** speedup over Continuum across diverse base models and datasets. This gain rises from our program-aware scheduler, which maintains a near-optimal KV cache hit rate (≈100% for Mini-SWE-Bench and OpenHands, see Figure 5 a, b, d, e) and enables the asynchronous preparation of environments. In contrast, Continuum suffers from performance degradation under high concurrency. As shown in Figure 5, its KV cache hit rate drops significantly from >90% to ≈60%. This is because Continuum suffers from KV cache eviction among requests in different programs

when no enough memory is available for ongoing requests' decoding. As a result, active programs compete for limited memory and trigger thrashing.

**Robustness performance to high concurrency.** THUN-DERAGENT maintains maximum achievable throughput even as the parallel workflow number scales beyond the GPU memory limit. As shown in Figure 4, THUNDER-AGENT ensures that throughput remains stable with the number of parallel workflows, whereas baseline systems suffer from severe throughput collapse once the workload exceeds memory limits. In practice, determining the optimal parallel workflow number to maximize utilization with limited KV cache thrashing and caching cost is infeasible due to the stochastic nature of agent environments and tool execution durations. THUNDERAGENT addresses this by automatically adapting to the maximum available capacity without manual tuning, a capability critical for robust real-world deployments.

**Robustness across deterministic and stochastic tool executions.** THUNDERAGENT outperforms baselines not only in workflows with deterministic tool patterns (Figure 4 a, b, d, e) but also under highly stochastic conditions (Figure 4 c, f). This comes from our dynamic program-aware waiting queue policy. vLLM's request-aware scheduler typically lacks reserved memory for acting programs, forcing frequent re-computation. Conversely, Continuum statically reserves memory for all paused programs and mispredicts the tool execution time. These lead to expensive $Cost_{recompute}$ or $Cost_{caching}$ during long, unpredictable tool calls. THUNDERAGENT balances them via a time-decay function $f(t)$, which prioritizes retaining KV cache for programs with short tool calls while preemptively pausing programs with long tool execution time to prevent memory

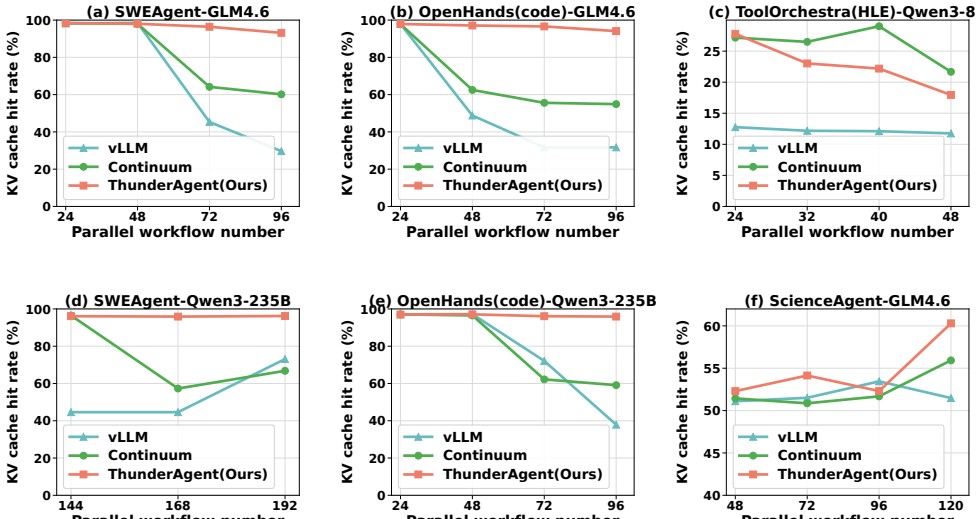

*Figure 5.* **KV Cache Hit Rate.** THUNDERAGENT achieves near-optimal KV hit rate with predictable tool call time (a, b, d, e), while dynamically trading KV cache hit rate for better compute utility when serving workflows with stochastic tool execution time (c, f).

*Table 2.* GLM-4.6 rollout ($N = 144$) on 2×H100 nodes.

| Workflow | Serving System | Throughput |
|---|---|---|
| mini-SWEAgent | vLLM + Gateway | 375.4 |
| mini-SWEAgent | THUNDERAGENT | 671.8 (**1.79**×) |
| OpenHands | vLLM + Gateway | 69.1 |
| OpenHands | THUNDERAGENT | 270.8 (**3.92**×) |

waste. As shown in Figure 5 (right), although THUNDERA-GENT exhibits a lower KV cache hit rate than Continuum in stochastic settings, it achieves higher throughput by ensuring active GPU utilization.

### 5.3. Rollout Evaluation Results
We evaluate RL rollout using GLM-4.6 on a two-node H100 cluster. Table 2 shows that THUNDERAGENT can maintain effective scalability, achieving a **1.79–3.92**× throughput increase over the vLLM + Gateway baseline, making it highly efficient for memory-intensive distributed RL workloads.

### 5.4. Ablation Study
**End-to-end latency breakdown.** Figure 6a decomposes the average end-to-end latency for OpenHands rollouts. The throughput gain stems primarily from reductions in **prefill and decode latency**. Moreover, the tool resource management policy (Section 4.4) contributes approximately **10%** to the latency improvement while providing **4.2**× disk memory savings. Per-step end-to-end latency are further discussed in Appendix E.

**Ablation on $\Delta t$ and f(t).** We study the sensitivity of detecting period $\Delta t$ and decaying function $f(t) = x^{-t}$. Figure 6b shows THUNDERAGENT offline serving mini-SWEAgent with GLM4.6 as base model on a single H100 node. We observe that THUNDERAGENT maintains high throughput under different parameter settings, demonstrating the robustness of our method. Further increasing $\Delta t$

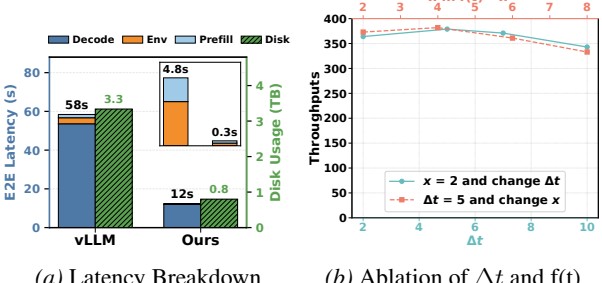

*(a)* Latency Breakdown     *(b)* Ablation of $\Delta t$ and f(t)

*Figure 6.* Ablation study of end-to-end latency breakdown and parameter sensitivity of THUNDERAGENT.

might decrease the KV cache hit rate and thereby reduce throughput because thrashing might occur in the middle of detecting. Also, increasing $x$ in $f(t)$ allows more aggressive eviction of acting programs, which trade recomputation costs to reduce caching costs. This reduces throughput as acting programs with short tool execution time are prematurely evicted.

## 6. Conclusion

We introduce THUNDERAGENT, a fast and simple agentic system built on a program-level abstraction that tracks metadata throughout the entire lifecycle of each agentic workflow. THUNDERAGENT leverages the program abstraction for runtime scheduling and resource management. Specifically, THUNDERAGENT dynamically schedules program execution across GPU nodes to mitigate KV cache thrashing and memory imbalance, while managing tool resources to prevent resource leakage. Experimental results showcase that THUNDERAGENT outperforms previous systems by **1.48–3.58**× for serving and **1.79–3.92**× for RL rollouts.

## Acknowledgements

We are grateful to Together AI for making this work possible. We thank Ben Athiwaratkun and Ce Zhang for assistance in developing the multi-backend scheduler. We thank Wenyi Hong and Luke Huang for helpful feedback and discussions during this work.

## Impact Statement

This paper presents work whose goal is to advance the field of Machine Learning Systems, specifically by optimizing the execution efficiency of agentic workflows. By significantly reducing the memory footprint and hardware requirements for maintaining agent inference and RL rollout, our method improves the **cost-efficiency** and **energy sustainability** of large-scale agent serving.

Our approach enables the training and inference of larger models and more complex environments on limited hardware resources. This efficiency helps companies and individuals to advanced agent research, allowing a broader range of institutions and practitioners to engage in high-fidelity simulations without necessitating prohibitive computational investments. While increasing system throughput could theoretically accelerate both positive and negative applications of autonomous agents, our work primarily targets infrastructure optimization, which is essential for the scalable and sustainable development of the field.

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

# A. Extended Discussion

## A.1. KV Cache Optimization

**Multi-tiered KV cache management.** To alleviate GPU memory pressure, systems such as Pensieve (Yu et al., 2025), Continuum (Li et al., 2025), Strata (Xie et al., 2025), and ShadowKV (Sun et al., 2025) exploit the hardware memory hierarchy, comprising GPU HBM, CPU DRAM, and NVMe SSD for KV cache management. These tiered caching mechanisms mitigate transient preemption by offloading inactive KV states to lower-tier storage and prefetching them back to GPU upon request resumption. However, the practical efficiency of these methods is fundamentally constrained by the inter-tier bandwidth between device and host memory. In high-frequency agentic workflows, the overhead of frequent swap-in and swap-out cycles often negates the benefits of multi-tier caching as shown in Section 5.

**Distributed KV cache management.** Distributed management of agentic states introduces significant complexity to KV cache eviction and preemption policies. While systems like BanaServe (He et al., 2025) and LMCache (Liu et al., 2025) enable KV cache transfer across DP nodes, their performance in large-batch agentic serving and rollout is often constrained by limited interconnect bandwidth. The strong intra-program dependencies in agentic workflows necessitate frequent state transferring without program-level management, which can easily saturate the network during serving or rollout.

To bypass these bandwidth bottlenecks, standard inference systems like vLLM KV-aware router (vLLM Team, 2025) and SGLang Model Gateway (Zheng et al., 2024) employ KV-aware routing policies that pin requests to specific nodes based on prefix locality or session ID. Similarly, Vortex (Yuan et al., 2025) introduces session-aware prefetching to minimize cross-node data transfer latency. However, these approaches lack the capability to dynamically migrate active program states between DP nodes. This absence of workload transfer leads to severe memory utilization imbalance across the cluster, shown in Figure 2a. Nodes hosting long-running agentic programs cannot offload states to idle peers, resulting in fragmented resource utilization and degraded aggregate throughput.

## A.2. Existing KV cache optimization methods

**KV cache offloading.** We investigated KV cache offloading by using LMcache (Liu et al., 2025) as a potential remedy for capacity constraints. While offloading theoritically extends effective memory space by utilizing CPU or SSD storage, our implementation with vLLM + LMcache reveals a critical bottleneck: the PCIe bandwidth is insufficient to sustain the high-frequency context switching and large-volume data transfers inherent to agentic workloads. As demonstrated in Figure 7a, when serving the GLM-4.6 model (Team et al., 2025) with the mini-SWEAgent framework (Yang et al., 2024), the latency penalty from frequent swap-in and swap-out operations negates the memory capacity benefits, resulting in severe throughput degradation under heavy agentic workloads.

**Prefill-Decode (PD) disaggregation.** We also explored PD disaggregation (Zhong et al., 2024), a standard optimization for chatbot serving by isolating the decoding phase from prefill interference. However, when applied to agentic workloads characterized by continuous context growth, we observe that PD disaggregation *exacerbates* thrashing. By partitioning the cluster into prefill-only and decode-only nodes, the effective HBM pool available for handling prefill is significantly smaller than that in a unified architecture. This memory fragmentation causes the system to hit capacity limits and trigger thrashing at much lower concurrency levels, as shown in Figure 7b. These results demonstrate that generic architectural optimizations cannot substitute for a program-centric scheduler that actively manages the working set.

## A.3. Scaling up agentic workflows

**Heterogeneous resource allocation and scheduling.** To orchestrate multi-turn agent-environment interactions at scale, recent systems such as MegaFlow (Zhang et al., 2026), RollArt (Gao et al., 2025; Wang et al., 2025a), AgentRL (Zhang et al., 2025), and VerlTool (Jiang et al., 2025) decouple model inference from environment execution. While these frameworks effectively scale environment concurrency via specialized services, they exhibit the inherent limitations of coarse-grained disaggregation. By treating the inference engine and tool executor as isolated black boxes, these systems lack unified resource management and are unable to coordinate KV cache lifecycles with the environment execution. Without fine-grained scheduling at program-level, disaggregation-based approaches waste KV cache reuse potential in agentic workloads, yielding sub-optimal throughput.

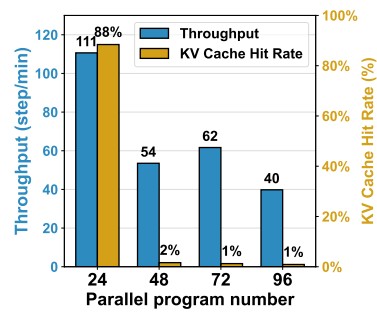
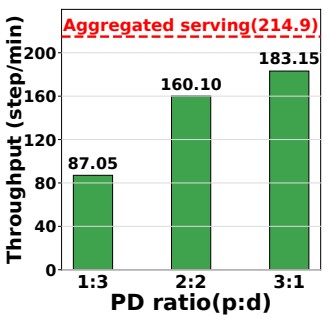

*(a)* KV Cache Hit Rate with KV Cache Offloading      *(b)* Throughput v.s. Prefill-Only/Decode-Only Node Ratio

*Figure 7.* Ablation study on KV cache offloading and Prefill-Decode (PD) disaggregation

**Scalability of THUNDERAGENT as a Routing Policy** For multi-node deployment, ThunderAgent replaces session-based static node-pinning with a global waiting queue. When a paused workflow is ready to resume, ThunderAgent routes its requests to the node with the most available capacity. Such routing strategy balances the tradeoff between KV cache locality and multi-node workload balance, achieving both efficient cache utilization and even load distribution across the cluster.

The main paper's distributed-rollout result (Table 2) uses the standard 2-node setting; we now extend the experiment to up to 8 nodes (64×H100) on the same GLM-4.6 + mini-SWEAgent workload, with concurrency scaled proportionally (72 programs per node). The baseline is the *Cache-Aware Routing Policy* in SGLang Model Gateway[2], which pins requests to specific backend replicas based on prefix locality but cannot redistribute paused programs across nodes.

Figure 8 compares throughput against this baseline and against an idealized linear-scaling reference. THUNDERAGENT tracks the linear-scaling line closely from 2 to 8 nodes, whereas the Cache-Aware Routing Policy quickly falls off because cross-node memory imbalance grows with cluster size and a locality-only router cannot rebalance the paused-program pool (cf. Section G.1). Concretely, THUNDERAGENT's speedup over the SGLang baseline *widens* from $1.79\times$ at 2 nodes to $2.39\times$ at 8 nodes. We also instrumented the per-tick communication delay of THUNDERAGENT's global scheduler: it grows sub-linearly from 13.5 ms at 2 nodes to 22.7 ms at 8 nodes (i.e., $1.68\times$ for a $4\times$ increase in nodes) and remains well below per-step inference latency, which is on the order of seconds. The global scheduler is therefore not a bottleneck for scaling.

### A.4. Compatibility of THUNDERAGENT with KV-cache offloading

THUNDERAGENT is orthogonal to KV-cache offloading and remains effective when offloading is enabled. To demonstrate this, we integrate THUNDERAGENT into **SGLang HiCache** (Xie et al., 2025) with the host-memory ratio set to 2.0, serving **MiniMax M2.5** on **R2E-Gym** across 8×H100 GPUs. We sweep serving batch size from 64 to 192 and compare against the same SGLang + HiCache stack *without* THUNDERAGENT as the baseline.

**Offloading alone is insufficient at high concurrency.** Figure 9 shows that the baseline keeps a high throughput up to batch size 96, where the combined host + device capacity still accommodates the working set. As we push to batch sizes 160 and 192, however, the baseline's combined capacity is exhausted: P50 latency jumps from $\sim$10 s to $\sim$30 s at batch 160 and to $\sim$70 s at batch 192, while throughput *decreases* despite the larger batch.

**THUNDERAGENT prevents capacity collapse and improves throughput at every batch size.** By temporarily pausing short programs to keep the live working set within available capacity, THUNDERAGENT holds both P50 and P90 latency essentially flat at $\sim$10 s and $\sim$20 s respectively across the entire sweep, while throughput grows monotonically with batch size. Even at small batch sizes where both methods achieve high cache hit via PCIe swapping, THUNDERAGENT still improves throughput and latency by selectively pausing acting programs.

### A.5. Portability of THUNDERAGENT across Hardware Generations

We test THUNDERAGENT on hardware with a lower compute-to-bandwidth ratio than H100. Table 3 reports the relevant ratio for the two GPU tiers we use in the paper: the A100 sits at 153 GFLOPS/GB, roughly half H100's 295.

---

[2]https://docs.sglang.io/docs/advanced_features/sgl_model_gateway#cache-aware-policy-tuning

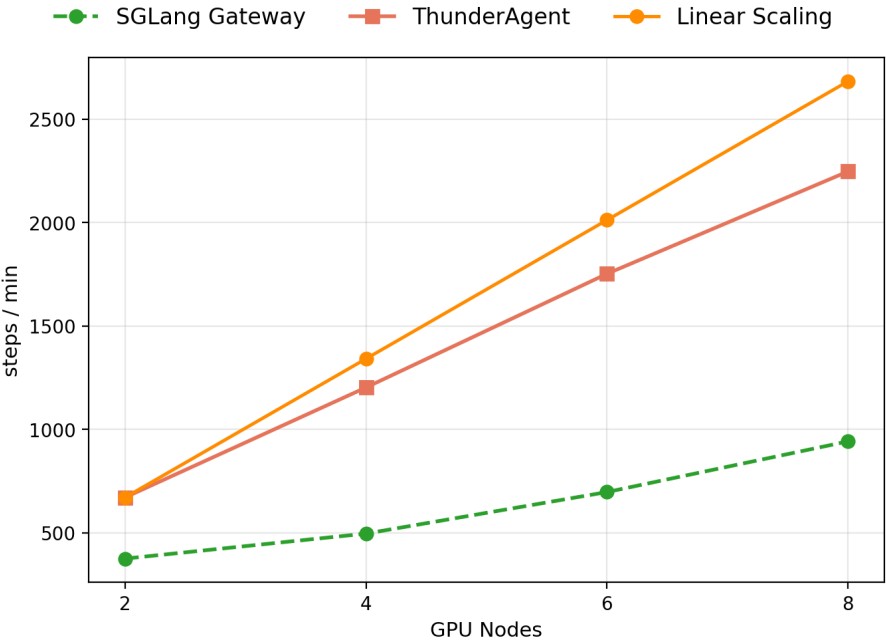

*Figure 8.* **Scaling up THUNDERAGENT to** 8 **H100 nodes**, GLM-4.6 + mini-SWEAgent (72 concurrent programs per node). THUNDER-AGENT tracks the ideal linear-scaling reference, while the Cache-Aware Routing Policy in SGLang Model Gateway falls progressively further behind as the cluster grows.

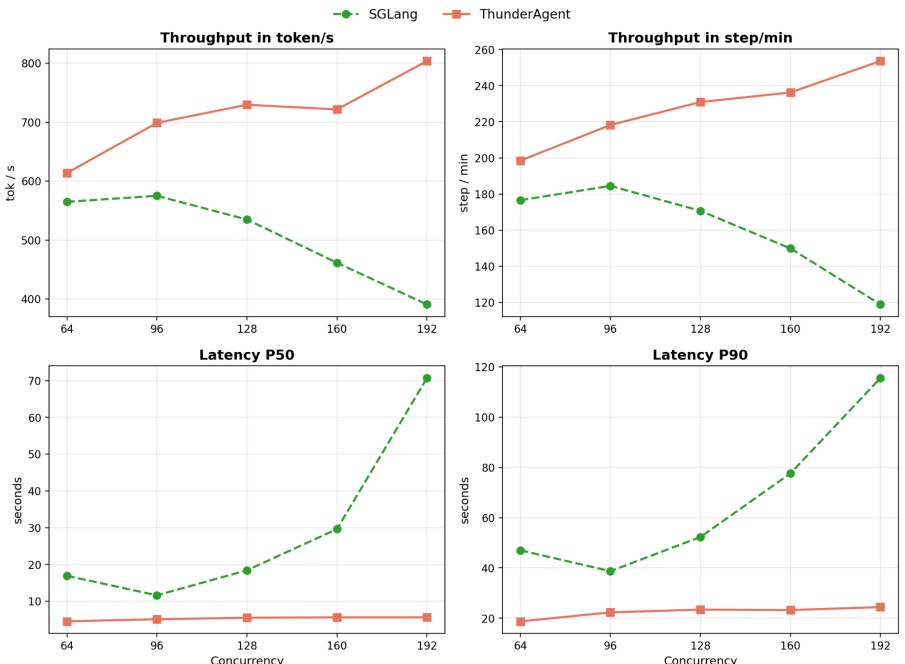

*Figure 9.* THUNDERAGENT with SGLang HiCache

Figure 10 reports throughput on 8×A100 with the same workloads as Figure 4(a, b). At low concurrency (24), THUNDERA-GENT matches vLLM within noise on both pipelines, because the working set fits comfortably in HBM and thrashing is rare. As concurrency rises to 48 and 72, the picture flips: THUNDERAGENT widens the gap to 1.71–2.08× on mini-SWEAgent and 1.16–1.62× on OpenHands, while vLLM's throughput *decreases* from concurrency 48 onward as it begins to thrash.

| GPU | Compute (TFLOPs) | Bandwidth (GB/s) | Ratio |
|-----|------------------|------------------|-------|
| H100 | 989 | 3350 | 295.2 |
| A100 | 312 | 2039 | 153.0 |

*Table 3.* Compute-to-bandwidth ratio across hardware tiers. We report FP16 Tensor Core (without sparsity) compute performance in TFLOPs and Global-to-L2 memory bandwidth in GB/s.

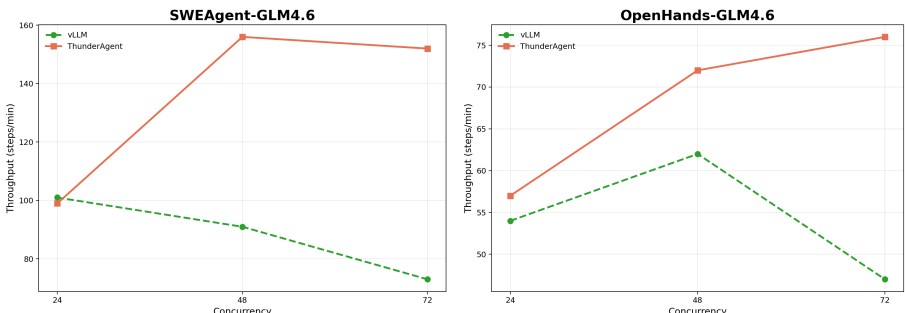

*Figure 10.* THUNDERAGENT on A100 GPUs

### A.6. Adoption of THUNDERAGENT in Open-Source Frameworks

Beyond our own evaluation, THUNDERAGENT has been adopted by open-source RL and serving frameworks to accelerate agentic rollout and serving, demonstrating that it can be integrated into existing stacks with small interface changes.

**SkyRL (RL rollout).** ThunderAgent is integrated into upstream SkyRL (Cao et al., 2025)'s main repository under `examples/train/thunder_agent` with an example training recipe at: `https://github.com/NovaSky-AI/SkyRL/tree/main/examples/train/thunder_agent`. This recipe uses THUNDERAGENT as a rollout gateway for SkyRL training. Each agent trajectory is exposed to THUNDERAGENT as a program through a `program_id`, and the program is explicitly released after termination, matching the minimal interface described in Appendix B. The released recipe trains Qwen3-32B on R2EGym with SkyRL's fully asynchronous framework and achieves a $3.01\times$ wall-clock speedup over the baseline, demonstrating that THUNDERAGENT can be adopted in an existing agentic RL stack with minimal interface changes.

**NVIDIA Dynamo (serving).** As part of Dynamo 2.0[3], THUNDERAGENT is being integrated into NVIDIA Dynamo[4], a datacenter-scale distributed inference serving framework, to improve throughput for agentic inference workloads.

---

[3] `https://github.com/ai-dynamo/dynamo/issues/9208`
[4] `https://github.com/ai-dynamo/dynamo/pull/9448`

# B. System Portability and Interface Abstraction.

## B.1. Middleware Architecture and Unified Interfaces.

THUNDERAGENT serves as **a program-aware runtime layer** that mediates between agent control flow and backend inference engines via a program-level abstraction. The scheduler controls program state transitions based on the abstracted `ProgramState` (see Table 4a) together with the backend cache capacity view (see Table 5). Meanwhile, each program binds only to the endpoint and does not depend on the concrete backend implementation.

## B.2. Why program ID matters.

While the standard **session ID** serves as a routing label, **the program ID is used by our system to check the workflow metadata.** This visibility is critical: it allows the scheduler to distinguish valid tool-wait times from idle sessions, enabling smart preemption strategies that session-based baselines cannot support.

**Only inference backend (e.g., vLLM/SGLang)**

```
1  # 1) LLM request
2  chat_completion(model_id, messages,
       extrabody)
3
4  # 2) Tool execution
5  run_tool(command, sandbox)
6
7  # 3) Program end
8  # (no explicit release)
```

**With ThunderAgent**

```
1  # 1) LLM request
2  extrabody["program_id"] = "PID"
3  chat_completion(model_id, messages,
       extrabody)
4
5  # 2) Tool execution
6  run_tool(command, sandbox,
       program_id="PID")
7
8  # 3) Program end (explicit release)
9  POST /programs/release
10 { "program_id": "PID" }
```

*Figure 11.* **Only three changes are required to use the ThunderAgent.**

## B.3. Low-overhead adoption of the ThunderAgent.

Figure 11 shows that adopting ThunderAgent **only requires** attaching `program_id` to requests (for both LLM inference and tool execution) and sending an explicit release signal with `program_id` when a program ends. The `program_id` tags each request with its own program instance for scheduling, while the release signal allows ThunderAgent to reclaim per-program resources after termination. **All other request fields and the OpenAI-style API surface remain unchanged.**

| Field | Type | Meaning |
|---|---|---|
| **ProgramState** | | |
| status | ProgramStatus | Current lifecycle state. |
| backend_url | str | Assigned backend endpoint. |
| step_count | int | Executed steps so far. |
| total_tokens | int | Total tokens over full history. |

*(a)* ProgramState fields.

| Status | Meaning |
|---|---|
| **ProgramStatus** | |
| REASONING | On-GPU inference. |
| ACTING | Off-GPU tool exec. |
| PAUSED | In global paused waiting set. |
| STOPPED | Released; resources reclaimed. |

*(b)* ProgramStatus semantics.

*Table 4.* Program state and status definitions.

| Field | Type | Meaning |
|---|---|---|
| **BackendState** | | |
| `url` | str | Backend endpoint. |
| `healthy` | bool | Health flag for scheduling. |
| `cache_config` | Optional[CacheConfig] | Static cache configuration (fetched at startup). |
| `active_program_tokens` | int | Active token footprint on this backend. |

*Table 5.* Key fields of BackendState.

## C. Tool execution time variability.

**Practical agent tool calls are hard to characterize and often unpredictable.** In some code-centric settings (e.g., serving SWE-Bench (Jimenez et al., 2024) with SWE-agent (Yang et al., 2024) or OpenHands (Wang et al., 2025b)), agents primarily invoke local, lightweight tools, and tool latency is relatively stable with low variance. However, in broader and more realistic scenarios, e.g., serving HLE (Phan et al., 2025) with ToolOrchestra (Su et al., 2025), the workload relies more heavily on remote-service tools (Table 6), making tool execution time volatile and difficult to predict. This volatility largely stems from factors external to the agent runtime, such as network jitter, backend load and queuing delays, and rate limiting, which can vary across requests and over time.

We empirically confirm this behavior in Figure 12. For remote-service tools (and some execution tools), the gap between the median and tail quantiles is large: p95 and p99 are substantially higher than the median, and the tail can extend to tens or even hundreds of seconds. This suggests that tool latency in these settings lacks a stable central tendency; instead, heavy-tailed behavior dominates, **making tool latency prediction intrinsically brittle in practice.**

Given the unpredictability of tool execution, underestimation wastes pinned cache capacity while still triggering premature KV eviction, causing thrashing upon resume. Overestimation, in contrast, may lead to unnecessary eviction of programs' KV that should have remained pinned. Even if tool runtimes were perfectly predictable, existing methods such as continuum (Li et al., 2025) still decide whether to keep the KV cache pinned using a static, threshold-based rule. In contrast, ThunderAgent builds **a complete cost-modeling framework** and **dynamically trades off** $\text{Cost}_{\text{recompute}}$ and $\text{Cost}_{\text{caching}}$.

| Tool bucket | Role | Primary variability source |
|---|---|---|
| `HLE-search` | Retrieve evidence | Remote service(Network latency/Rate limits) |
| `HLE-enhance-reasoning` | Model-as-a-tool call | Remote service |
| `HLE-answer` | Final generation | Local LLM inference |
| `SAB-execute_bash` | Shell execution | Sandbox and I/O |
| `SAB-execute_ipython_cell` | Python cell execution | Program runtime |
| `SAB-str_replace_editor` | File edit | Local filesystem |
| `SAB-task_tracker` | Task state tracking | Local filesystem |

*Table 6.* Tool buckets.

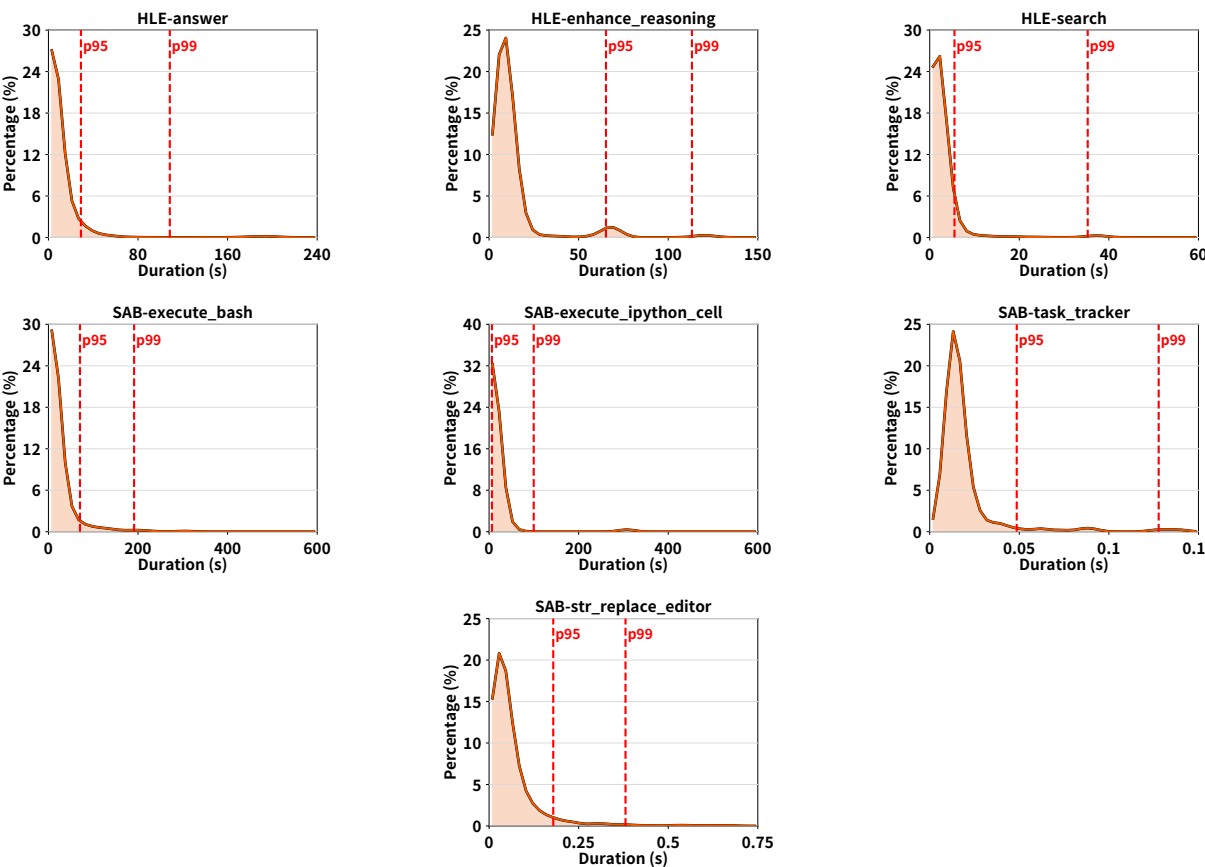

*Figure 12.* **Tool execution time distributions.** Tool execution time exhibits high variability and is difficult to predict.

## D. KV cache hit rate statistics and interpretation

In our cost decomposition Equation (3), throughput loss in agentic serving mainly comes from *non-productive* overheads: KV re-computation induced by thrashing and idle KV caching during external tool execution, i.e., $Cost_{recompute}$ and $Cost_{caching}$. When tool calls are short and predictable, the acting phase occupies KV for only a short time, so $Cost_{caching}$ is small; thus, avoiding thrashing dominates: a higher KV cache hit rate typically implies fewer re-prefills and higher throughput.

However, when tool execution times are highly variable (see Appendix C), a TTL-based scheduler can end up pinning the KV for long tool calls. While this can reduce $Cost_{recompute}$ and thus increase the KV cache hit rate, it simultaneously inflates $Cost_{caching}$ and reduces throughput. This helps explain why continuum (Li et al., 2025) can underperform on tool-heavy workloads despite achieving a higher KV cache hit rate (Figs. 4, 5).

**ThunderAgent adapts to these regimes by explicitly balancing caching and recomputation.** ThunderAgent introduces a time-decay function $f(t)$ in Sec. 4.3 for acting programs to trade off $Cost_{caching}$ and $Cost_{recompute}$; we rigorously derive the optimal functional form of $f(t)$ in Appendix F.1. By progressively lowering the effective memory priority of long-idle acting programs, the scheduler evicts their KV caches to reduce idle caching cost while controlling recomputation, yielding better throughput in practice (Figs. 4).

## E. End-to-End Latency Analysis

Though we have stated in Section 1 that program-level latency(time used for whole workflow generation) is far more important than end-to-end per step latency for autonomous agents and agentic RL rollout. Here we compare THUNDERAGENT's average per-step latency with vLLM and Continuum. Figure 13 shows that THUNDERAGENT significantly outperforms vLLM and Continuum when applying GLM4.6 and Qwen3 235B with mini-SWEAgent and Openhands on a single H100 serving in either low or high parallel workflow number. **The reason is that it seems to improve end-to-end latency by switching acting programs. But it actually delays all the running programs' latency by triggering heavy KV-cache thrashing.**

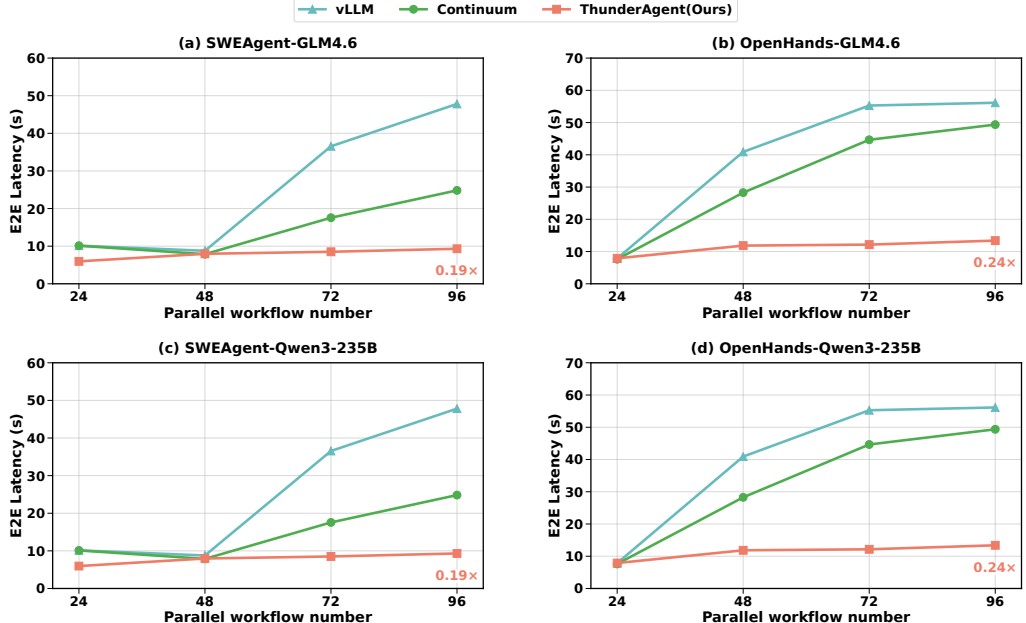

*Figure 13.* End-to-End latency comparision

# F. Extended Theoretical Analysis.

## F.1. Proof of Time Decay Function for Periodic Thrashing Detection.

**Hypothesis F.1** (Unpredictable Tool Execution Time). For acting programs, we hypothesize that the scheduler cannot reliably predict the tool return time for a given program (see Appendix C). Consequently, the decay function $f$ should depend only on the elapsed acting time $t$ in a time-homogeneous manner (Parzen, 1999).

**Hypothesis F.2** (Boundary Conditions). We assume the time decay function $f : [0, \infty) \to (0, 1]$ satisfies

$$f(0) = 1, \lim_{t \to \infty} f(t) = 0, \tag{12}$$

An intuitive interpretation of these boundary conditions is that, when the tool execution time is 0, corresponding to a multi-turn interaction without tool calls, all acting programs reduce to reasoning programs, and therefore $f(t) = 1$. Conversely, if the tool execution time is infinite, the agentic workflow collapses to single-turn generation, akin to standard chatbot serving, since requests never return for the next-turn interactions. In this regime, setting $f(t) = 0$ aligns the decay function with request-level scheduling policies.

**Theorem F.3** (Admissible Time Decay Functions). *Under Hypothesis F.1 and F.2, the admissible time decay function $f$ for our capacity check function in Equation 7 must take one of the following forms: exponential in continuous time, $f(t) = e^{-\lambda t}$ with $\lambda > 0$, or geometric in discrete tick time, $f(k) = x^{-k}$ with $x > 1$.*

*Proof.* We prove this theorem by first formalizing the time-homogeneous property implied by Hypothesis F.1. Next, we inducing the admissible time decay functions $f$ under the boundary conditions in Hypothesis F.2.

**Formalization of unpredictable tool time.** Let $t$ denote the elapsed acting time, measured in wall-clock time (continuous time) or in periodic-monitor ticks (discrete time). Under Hypothesis F.1, the relative decay after waiting an additional duration $\Delta$ should not depend on the absolute elapsed time $t$, but only on the increment $\Delta$. We formalize this as the existence of a function $\phi : [0, \infty) \to (0, 1]$ such that, for all $t, \Delta \geq 0$,

$$f(t + \Delta) = f(t) \, \phi(\Delta). \tag{13}$$

**Semigroup equation.** Setting $t = 0$ in Equation 13 and using the boundary condition $f(0) = 1$ (from Hypothesis F.2) yields $\phi(\Delta) = f(\Delta)$. Substituting back, we obtain the multiplicative semigroup equation

$$f(t + \Delta) = f(t) \, f(\Delta), \qquad \forall t, \Delta \geq 0. \tag{14}$$

**Continuous-time case (exponential decay).** We first consider the continuous-time case. Define $h(t) \triangleq \ln f(t)$. Applying the logarithms on both sides of Equation 14 yields the *Cauchy functional equation*

$$h(t + \Delta) = h(t) + h(\Delta). \tag{15}$$

Since $f(t) \in (0, 1]$, we have $h(t) \leq 0$ for all $t \geq 0$, which implies that $h$ is bounded above on $[0, \infty)$. Under this boundedness condition, the Cauchy functional equation admits only linear solutions of the form $h(t) = ct$ for some $c \in \mathbb{R}$. Writing $\lambda \triangleq -c \geq 0$, we obtain

$$f(t) = e^{-\lambda t}. \tag{16}$$

Finally, the boundary condition $\lim_{t \to \infty} f(t) = 0$ (Hypothesis F.2) rules out $\lambda = 0$, and thus $\lambda > 0$.

**Discrete-time case (geometric decay).** We next consider the discrete-time setting, where elapsed acting time is measured in integer ticks $k \in \mathbb{Z}_{\geq 0}$. Equation 14 becomes

$$f(m + n) = f(m) \, f(n), \qquad \forall m, n \in \mathbb{Z}_{\geq 0}. \tag{17}$$

Setting $n = 1$ yields the recurrence $f(k) = f(k-1)f(1)$. Let $\gamma \triangleq f(1)$, we have $f(k) = f(1)^k \triangleq \gamma^k$. The boundary condition $\lim_{k \to \infty} f(k) = 0$ implies $0 < \gamma < 1$. Equivalently, we can parameterize

$$f(k) = x^{-k}, \qquad x \triangleq \gamma^{-1} > 1. \tag{18}$$

This completes the proof. $\qquad\qquad\square$

## F.2. Proof of recomputation STP cost

As defined in Section 4.2, the STP recomputation cost is given by:

$$\text{Cost}_{\text{recompute}} = \int_0^{t_{recompute}} c_i(t) \, dt \tag{19}$$

where $c_i(t)$ represents the instantaneous cost, which is proportional to the decoding step (i.e., $c_i(t) \propto t$). This proportionality arises because chunked prefill processes a constant number of KV pairs per iteration, resulting in a linear increase in accumulated computation over time. Consequently, evaluating the integral yields $\text{Cost}_{\text{recompute}} \propto t_{\text{recompute}}^2$. Given the relationship $t_{\text{recompute}} = c_i \times T_{\text{decode}}/\text{chunk}$, where both $T_{\text{decode}}$ and the chunk size are constant, it follows that:

$$Cost_{\text{recompute}} \propto c_i^2$$

## F.3. Proof of minimized recomputation STP cost

We provide a rigorous proof for the optimality of the Shortest-First Eviction policy using an **exchange argument**.

**Problem Definition.** We aim to select a subset of paused programs $S$ to evict such that the total reclaimed memory satisfies $\sum_{i \in S} c_i \geq \Delta C$, while minimizing the total re-computation cost $J(S) = \sum_{i \in S} c_i^2$. Note that the cost function $f(x) = x^2$ is strictly convex and super-additive (i.e., $(a + b)^2 > a^2 + b^2$ for positive $a, b$).

**Theorem.** The optimal strategy to minimize $J(S)$ is to strictly select programs with the smallest context lengths $c_i$.

**Proof.** Suppose, for the sake of contradiction, that the optimal set $S^*$ is *not* the set of the shortest programs. This implies there exists a "long" program $p_{long} \in S^*$ and a "short" program $p_{short} \notin S^*$ (available but not selected) such that $c_{short} < c_{long}$.

We can construct a new set $S'$ by swapping or decomposing $p_{long}$. Since $c_{long} > c_{short}$, we can conceptualize $p_{long}$ as being composed of a segment of length $c_{short}$ and a residue $r = c_{long} - c_{short}$.

Replacing the selection of $p_{long}$ with $p_{short}$ (and theoretically the residue $r$) changes the cost. Consider the inequality derived from the convexity of the square function:

$$c_{long}^2 = (c_{short} + r)^2 = c_{short}^2 + r^2 + 2c_{short}r \tag{20}$$

Since $c_{short} > 0$ and $r > 0$, the cross-term $2c_{short}r > 0$. Therefore:

$$c_{short}^2 + r^2 < c_{long}^2 \tag{21}$$

This inequality implies that breaking a large eviction target ($c_{long}$) into smaller components ($c_{short} + r$) strictly reduces the sum of squares. In the context of our scheduler, this means that if we are satisfying the memory constraint $\Delta C$ using a large program, we can strictly decrease the penalty by swapping it for available smaller programs (or a combination thereof) that sum to the same capacity.

By iteratively applying this exchange—replacing the largest selected programs with smaller unselected programs—we monotonically decrease the cost function $J(S)$. The cost reaches its global minimum only when no such exchange is possible, i.e., when $S$ consists entirely of the programs with the smallest available context lengths.

**Conclusion.** The Shortest-First strategy is globally optimal because the super-linear cost of attention ($O(L^2)$) penalizes fragmentation less than aggregation.

# G. Additional Ablation Studies

We provide additional ablation studies that isolate the contribution of THUNDERAGENT's individual design decisions, complementing the parameter-sensitivity ablation in Section 5.4.

## G.1. Global vs. Local Waiting Queue

We compare THUNDERAGENT's **global** waiting queue, which permits paused programs to be restored on a different DP node from their original backend, against a **local** waiting queue that always restores a program to its origin node. In the local variant, programs that pause on memory-saturated nodes must wait until their original node has capacity, whereas the global queue redistributes them to under-utilized peers.

As shown in Figure 14, the global queue consistently improves throughput as we scale from 2 to 4 H100 nodes. At 4 nodes, the gain widens to $1.28\times$, reflecting that cross-node memory imbalance grows with cluster size (Figure 2a); the global queue directly mitigates this imbalance.

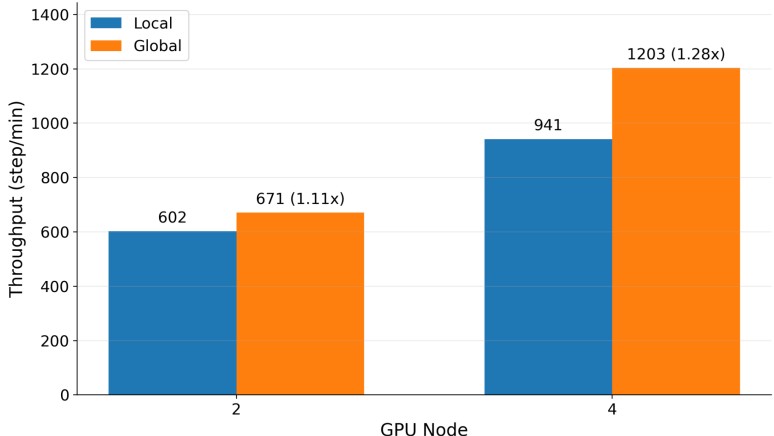

*Figure 14.* **Global vs. local waiting queue** on GLM-4.5-fp8 + mini-SWEAgent (8xH100 GPU, 72 concurrent programs per node).

## G.2. Time-Decay Function Form

Theorem F.3 establishes that exponential decay is the unique admissible form of $f(t)$ under the unpredictable-tool-latency assumption and the boundary conditions in Hypothesis F.2; we now empirically compare it against two practical alternatives: **constant** ($f(t) = 1$, i.e., no decay), **linear** ($f(t) = \max(1 - t, 0)$), and **exponential** ($f(t) = e^{-t}$). We evaluate on three agentic pipelines with progressively less predictable tool latency: mini-SWE-Agent (short, regular tool calls), OSWorld (longer but largely periodic tool calls, e.g., a fixed 5s interval between screenshots), and ScienceAgent (highly stochastic, heavy-tailed tool calls; see Appendix C).

As shown in Figure 15, exponential decay is the most robust choice overall: it outperforms both alternatives on mini-SWE-Agent ($1.08\times$) and ScienceAgent ($1.63\times$ vs. $1.39\times$ for linear), where unpredictable or long-tailed tool latency exposes the brittleness of a static $f(t)$. On OSWorld, however, exponential decay trails linear by about $6\%$ ($1.14\times$ vs. $1.21\times$): when tool durations are themselves near-deterministic, a linearly shrinking weight tracks the predictable tool clock more tightly. Constant decay, which never reclaims memory from long-idle acting programs, is the weakest on every workload that exhibits any waiting at all.

### G.3. Eviction Policy

We empirically validate the shortest-first eviction policy proved optimal in Section F.3 against two natural baselines: **random** eviction and **longest-first** eviction. We run on two agentic pipelines, mini-SWE-Agent (*SWE*) and OpenHands, both with GLM-4.5-fp8 on a single H100 node at 72 concurrent programs.

As predicted by the $O(L^2)$ recomputation cost (Section F.2), evicting the longest paused programs is the worst choice: the dropped KV state is the most expensive to rebuild on resume, inflating average prefill time and starving the decoding stream. Shortest-first achieves the highest throughput and the lowest average prefill time on both pipelines (Figure 16). The gap is largest on OpenHands, where program contexts are heaviest – shortest-first is roughly $2.0\times$ the throughput of longest-first and cuts average prefill time by $2\times$ (7.5 s vs. 15.1 s). Random falls between the two, consistent with it being an unbiased estimator over the same cost surface.

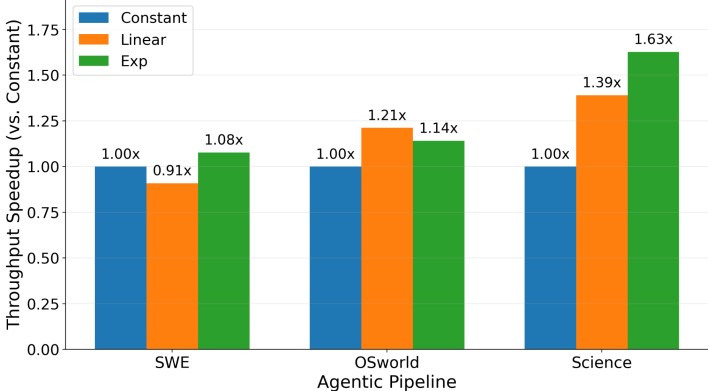

*Figure 15.* **Ablation of the decay function** $f(t)$ across three agentic pipelines. Throughput is reported as speedup vs. the constant (no decay) baseline. Exponential decay is the most robust overall; linear decay is preferable only on OSWorld, where tool execution time is near-deterministic. Experiments are conducted on an 8xH100 node with 72 concurrent programs using GLM-4.5-fp8 model.

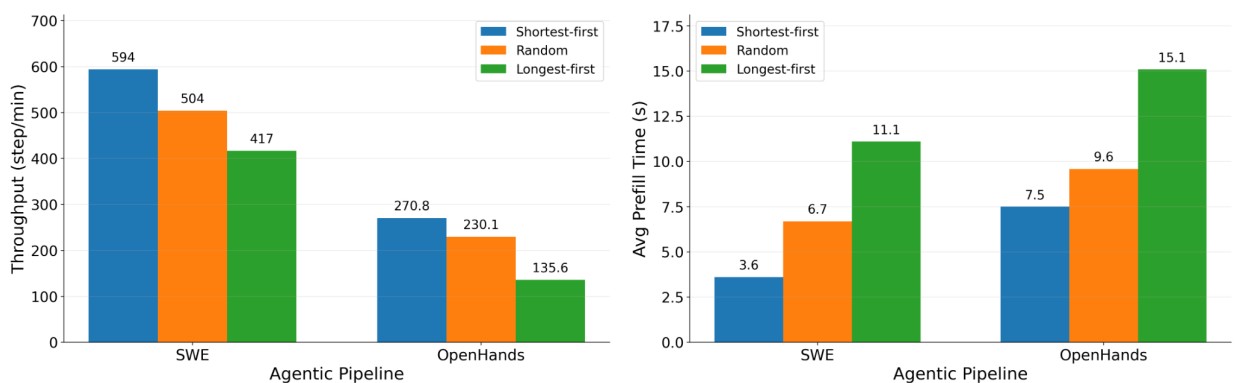

*Figure 16.* **Ablation of the eviction policy.** (Left) throughput, (right) average prefill time. Shortest-first eviction wins on both axes, with the gap widening on the heavier-context OpenHands pipeline. Experiments are conducted on an 8xH100 node with 72 concurrent programs using GLM-4.5-fp8 model.

## G.4. Component-wise Contribution

We disentangle the contribution of THUNDERAGENT's two main scheduling components by starting from vanilla vLLM and incrementally enabling **local program-aware scheduling** (the cost-driven eviction + decay function operating per backend) and then the **global waiting queue** (cross-node program migration). Table 7 reports throughput on $2\times$H100 nodes with GLM-4.5-fp8 + mini-SWEAgent at $144$ concurrent programs.

The largest jump comes from local program-aware scheduling, which raises throughput from $375$ to $602$ steps/min ($1.61\times$) by directly suppressing unnecessary recomputation and idle caching cost (Equation 3) within each backend. Layering the global waiting queue on top adds a further $1.12\times$ ($602 \rightarrow 672$) by reducing cross-node memory imbalance, consistent with the standalone global-vs-local result in Section G.1.

| Component | Throughput (steps/min) |
|---|---|
| vLLM (baseline) | 375 |
| + local program-aware scheduling | 602 |
| + global waiting queue (THUNDERAGENT) | 672 |

*Table 7.* Component-wise ablation on $2\times$H100 + GLM-4.5-fp8 + mini-SWEAgent at $144$ concurrent programs. Speedups are reported against vLLM. Local scheduling delivers the majority of the gain; the global queue is responsible for the remaining cross-node imbalance reduction.

