# OpenReview forum: "ThunderAgent: A Fast, Simple, and Program-Aware Agentic Inference System"
_ICML.cc/2026/Conference — ICML 2026 spotlight_

### Official Review · Reviewer_RukN · 2026-03-09

**Soundness:** 3
**Presentation:** 3
**Significance:** 3
**Originality:** 3
**Overall Recommendation:** 4
**Confidence:** 3

**Summary:**

This paper introduces ThunderAgent, an inference system for LLM agents that abstracts workflows as LLM Programs. It manages KV-caches and tool resources by prioritizing reasoning over acting phases. Key contributions include a shortest-context-first eviction policy and a global waiting queue for cross-node load balancing. Evaluations show 1.5-3.6x throughput gains in serving and up to 3.9x in RL rollouts.

**Compliance With Llm Reviewing Policy:**

Affirmed.

**Key Questions For Authors:**

1. In hardware with lower compute-to-bandwidth ratios (e.g., A10), does the recompute strategy still outperform KV-cache offloading?
2. Is there a specific context-length threshold where the recomputation penalty necessitates a return to saving or pinning strategies?
3. How does the average completion time of a single agent workflow change as parallel concurrency increases from 24 to 96?
4. How does the synchronization overhead of the global queue scale with more nodes?

**Limitations:**

Yes

**Strengths And Weaknesses:**

Strength:
+ ThunderAgent maintains superior throughput in stochastic tool settings where Continuum’s duration predictions often fail.
+ ThunderAgent’s global waiting queue resolves the peak memory imbalance observed in locality-aware routers. Continuum lacks a mechanism for such cross-node redistribution.
+ Unlike Continuum’s static TTL-based pinning, ThunderAgent uses a time-decay function to trade off caching and recomputation costs. This prevents memory underutilization during unpredictable tool calls.

Weakness:
- The "Evict & Recompute" strategy assumes re-prefilling on high-end GPUs is faster than PCIe data swapping. This may not hold for mid-range hardware with lower compute-to-bandwidth ratios.
-The re-prefilling is only cheap for short sequences. For agents with long context histories, even shortest-first eviction could cause significant stalls.
-  Increasing concurrency from 24 to 48 improves system throughput but likely increases individual task turnaround time due to global queuing.
-  The global scheduler (ticking every 5s) may become a communication bottleneck as the cluster scales beyond the 2-node setup.

---

> ### Author Rebuttal · Authors · 2026-03-30
>
> # W1 and Q1: Compatibility with KV cache offloading
> We’d like to clarify that:
>
> **(1) Orthogonality**: ThunderAgent is orthogonal to, and fully compatible with, KV-cache offloading. It does not assume re-prefilling is faster than PCIe swapping.
>
> **(2) Capacity Management**: ThunderAgent triggers eviction only when the total available KV cache capacity (including offloading tiers) is exhausted. With hierarchical storage enabled, it simply manages the expanded pool.
>
> **(3) Inevitability of Recomputation**: Offloading delays but does not eliminate capacity exhaustion. When working set exceeds all available storage, eviction becomes unavoidable.
>
> **(4) Dynamic Tradeoff**: Rather than a static "Evict & Recompute" strategy, ThunderAgent dynamically trades off evicting and preserving based on real-time token capacity across backends (Section 4.3).
>
> To answer your question: Yes, the strategy remains effective (a) on lower compute-to-bandwidth hardware and (b) when offloading is enabled. To demonstrate this, we provide additional experiments:
>
> **(a) results on A100**: We evaluate on 8×A100s (which have lower compute-to-bandwidth than A10), with settings identical to Figure 4(a)(b). As shown in https://anonymous.4open.science/r/anonymous-icml26-submission-B4EF/R4/R4T1.md, ThunderAgent achieves similar throughput improvements on A100.
>
> **(b) Offloading-enabled results**: We integrate ThunderAgent into SGLang+HiCache (host memory ratio=2.0), serving MiniMax M2.5 on R2E-Gym with 8×H100s. As shown in https://anonymous.4open.science/r/anonymous-icml26-submission-B4EF/R4/R4T2.md, offloading alone does not resolve capacity exhaustion: at batch size 192, the baseline exhausts combined host/device capacity, leading to low KV cache hit and throughput degradation. ThunderAgent prevents this by selectively pausing short programs to maintain high cache hit rates. Even at small batch sizes (64 to 128) where both methods achieve high cache hit via PCIe swapping, ThunderAgent still improves throughput and latency by selectively pausing acting programs.
> # W2 and Q2: Recomputation vs pinning for long contexts
> We’d like to clarify that:
>
> **(1) High Cost of Pinning**: While re-prefilling long sequences is expensive, pinning them is also highly costly, as holding a long KV cache during tool execution wastes GPU memory and reduces generation throughput, formalized as $\text{Cost}_\text{caching}$ in Equation 3.
>
> **(2) Unpredictable Tool Latency**: In downstream workloads where tool latencies are highly stochastic and heavy-tailed (Appendix C, Figure 8), predictive pinning strategies like Continuum frequently mispredict, causing severe memory underutilization and even worse performance than vLLM on ToolOrchestra and ScienceAgent (Figure 4 c,f).
>
> **(3) Dynamic Capacity Modeling**: ThunderAgent does not simply default to eviction. It monitors backend GPUs’ real-time KV capacity, dynamically adjusting the trade-off between preserving and evicting KV caches.
>
> To answer your question: there is no universal context-length threshold where pinning beats recomputation, because pinning cost depends on unpredictable tool execution times. Instead of relying on static thresholds, ThunderAgent makes tradeoffs dynamically: under high memory pressure, it pauses short programs to free generation capacity; under low pressure, it retains programs’ caches (which achieves similar results to pinning). This automatic trade-off is why ThunderAgent remains robust across diverse workloads.
> # W3 and Q3: Throughput vs latency tradeoff
> We’d like to clarify that:
>
> **(1) Fundamental Tradeoff**: The tradeoff between concurrency and per-request latency exists in all LLM inference systems, being bounded by finite GPU compute and memory rather than an artifact of ThunderAgent's design.
>
> **(2) Global Queuing Actually Mitigates Latency Degradation**: As shown in Figure 10, when batch size increases from 24 to 96, vLLM and Continuum suffer severe latency degradation from KV cache thrashing, while ThunderAgent maintains a nearly flat latency profile with increasingly higher throughput, expanding the Pareto frontier in throughput-latency tradeoff.
> We provide raw data for Figure 4(a), 9(a), and 10(a) in https://anonymous.4open.science/r/anonymous-icml26-submission-B4EF/R4/R4T3.md
> # W4 and Q4: Scalability and communication overhead
> We’d like to clarify that the global scheduler is lightweight by design: each asynchronous tick only queries per-backend aggregate token counts without transferring actual KV cache. We validate this scaling to 8 nodes (64 H100 GPUs) on the same workload as Figure 4(a), as shown in https://anonymous.4open.science/r/anonymous-icml26-submission-B4EF/R4/R4T4.md
> Communication delay grows sub-linearly from 13.5ms (2 nodes) to 22.7ms (8 nodes), which is negligible versus per-step latency (seconds-level). ThunderAgent's speedup over vLLM's KV-aware router widens at scale (from 1.79× at 2 nodes to 2.39× at 8 nodes), confirming the scheduler is not a bottleneck.

---

> > ### Author Rebuttal · Reviewer_RukN · 2026-04-02
> >
> > Thanks for the response. I will keep my score.

---

### Official Review · Reviewer_atjC · 2026-03-12

**Soundness:** 3
**Presentation:** 3
**Significance:** 3
**Originality:** 3
**Overall Recommendation:** 5
**Confidence:** 4

**Summary:**

When an AI agent is working on a task, it alternates between thinking (using the GPU) and waiting (for a tool like a code compiler to finish). During that waiting period, existing systems waste or lose the GPU's memory of what the agent was doing, so when the tool finishes, the system has to relearn the entire conversation from scratch. This gets very expensive at scale.
ThunderAgent fixes this by giving the inference system a bird's-eye view of the entire workflow instead of handling each LLM call in isolation. It smartly decides which agent's memory to keep, which to temporarily drop, and how to spread work across GPUs, all while preparing tool environments in the background before they're needed.

**Compliance With Llm Reviewing Policy:**

Affirmed.

**Final Justification:**

Thank you for the clarification on Figure 4(c) and for the honest limitations discussion. My concerns are fully resolved and I am raising my score accordingly.

**Key Questions For Authors:**

None

**Limitations:**

The paper doesn't discuss scenarios where ThunderAgent might perform worse than baselines, which is a notable omission for a research paper.

**Strengths And Weaknesses:**

Strengths:
ThunderAgent tackles an important and practical problem in agentic AI: the wasteful mismanagement of GPU memory during the idle periods when an agent is waiting for an external tool to finish. By maintaining a program-level view of the entire workflow. In terms of performance, it achieves 1.48–3.58× speedup for standard inference serving and 1.79–3.92× speedup for RL rollout, along with up to 4.2× savings in disk memory.

Weaknesses:
1. When tool execution times are highly unpredictable, the system's ability to make smart scheduling decisions is constrained.
2. Hyperparameters such as delta t and the decay function need to be configured per deployment

---

> ### Author Rebuttal · Authors · 2026-03-30
>
> # W1: Handling unpredictable tool latency and scheduling effectiveness
>
> The reviewer is right that high variability in tool execution times is challenging. However, this is precisely the scenario ThunderAgent is designed for.
>
> As discussed in Section 4.3.1, to reduce system performance instability caused by variable tool latency, we treat tool execution times as unpredictable and use an exponential decay factor to automatically trade off caching cost against recomputation cost; Appendix E.3 further proves that, under this unpredictability assumption, exponential decay is the unique admissible form.
>
> In practice, our approach demonstrates robust performance across diverse settings, as validated in Figure 4. In particular, ToolOrchestra on HLE invokes remote-service tools such as HLE-search, while OpenHands on ScienceAgentBench invokes execution tools such as SAB-execute ipython cell. As shown in Appendix C, these tools have large median–p95/p99 gaps, with tails extending to tens or even hundreds of seconds. Despite this high latency variability, the table below shows that ThunderAgent remains effective on both workloads.
>
> ToolOrchestra Throughput Data (steps/min)
>
> |Concurrency|24|32|40|48|
> |:-:|:-:|:-:|:-:|:-:|
> |vLLM|5.08|5.17|5.5|4.78|
> |Continuum|3.37|3.39|4.19|3.09|
> |ThunderAgent|6.05|6.46|6.95|7.06|
>
> ScienceAgent Throughput Data (steps/min)
>
> |Concurrency|48|72|96|120|
> |:-:|:-:|:-:|:-:|:-:|
> |vLLM|40.15|41.08|42.77|43.55|
> |Continuum|35.18|35.37|38.73|46.88|
> |ThunderAgent|41.61|47.15|48.38|51.5|
>
> # W2: Robustness of $\Delta t$ and the decay function in practice
>
> We appreciate the reviewer's concern. However, Δt and the decay function are not sensitive hyperparameters in practice.
>
> As shown in the tables above and Section 5.4, throughput remains high across a broad range of settings: ~344–379 when varying $\Delta t$ with $x = 2$, and ~334–382 when varying $x$ with $\Delta t = 5$. This indicates that ThunderAgent is robust to both $\Delta t$ and the decay function.
>
> Sensitivity to $\Delta t$ (with $x = 2$)
>
> | $\Delta t$ | 2 | 5 | 7 | 10 |
> |:-:|:-:|:-:|:-:|:-:|
> | Throughput | 364 | 379 | 371 | 344 |
>
> Sensitivity to $x$ in $f(t) = x^{-t}$ (with $\Delta t = 5$)
>
> | $x$ | 2 | 4 | 6 | 8 |
> |:-:|:-:|:-:|:-:|:-:|
> | Throughput | 374 | 382 | 361 | 334 |

---

> > ### Author Rebuttal · Reviewer_atjC · 2026-04-02
> >
> > Thanks for the detailed rebuttal. Both concerns are adequately addressed. The exponential decay derivation is a principled response to the unpredictable tool latency concern, not just an empirical claim, and the hyperparameter sensitivity tables show the system is robust across a reasonable range.
> > My remaining concern is the one I raised in my original review, scenarios where ThunderAgent underperforms baselines, which is still not discussed. Looking at Figure 4c, ThunderAgent appears to lose to Continuum at low batch sizes on the ToolOrchestra workload, and this is never acknowledged as a failure mode. A systems paper should be transparent about its operating envelope. If the authors can include a brief, honest discussion of these boundary conditions in the final version, where the system works well, and where it doesn't, I would be willing to raise my score.

---

> > > ### Author Response · Authors · 2026-04-02
> > >
> > > Thank you for this helpful suggestion. We agree that the paper should clearly state ThunderAgent’s operating envelope and boundary conditions. We appreciate this suggestion and will add a short, explicit discussion of these boundary conditions in the final version.
> > > # ThunderAgent outperform Continuum in Figure4
> > >
> > > At low batch sizes in Fig. 4(c), ThunderAgent does **not** underperform Continuum: it remains above both Continuum and vLLM. We have included the raw data points in our response to Weakness 1 in the table titled “ToolOrchestra Throughput Data (steps/min)” at previous response. In Figure 4(c), ThunderAgent is represented by the **coral red line**, which is **above the green line representing Continuum** in any batch size.
> > >
> > > # Limitation of ThunderAgent
> > > ## Limited throughputs under low memory preasure.
> > >
> > > When the workload fits comfortably in available KV capacity, thrashing is rare, so ThunderAgent has less room to improve throughput. In such cases, its scheduling overhead can slightly reduce throughput. ThunderAgent is designed for multiturn react agent serving/rollout. So the design does not improve single turn question-answering chatbot. Instead, these additional communication overhead decrease throughputs within **10%** when serving GLM-4.5-fp8 on GSM8K at [R2T1-2](https://anonymous.4open.science/r/anonymous-icml26-submission-B4EF/R2/R2T1.md).
> > >
> > >
> > > ## Suboptimal for exponential decay
> > >
> > > Our default exponential decay is designed for unpredictable tool-return times and is the most robust overall, but it is not always best in every workload. In OSWorld, where tool calls are long and highly regular (e.g., fixed 5-second screenshot intervals), exponential decay is about 6% worse than linear decay, though still better than constant(f(t) = 1) shown in [R2T3](https://anonymous.4open.science/r/anonymous-icml26-submission-B4EF/R2/R2T3.png).
> > >
> > > We appreciate this suggestion and would be happy to clarify any remaining questions or concerns.

---

### Official Review · Reviewer_bPw6 · 2026-03-12

**Soundness:** 2
**Presentation:** 2
**Significance:** 3
**Originality:** 3
**Overall Recommendation:** 4
**Confidence:** 4

**Summary:**

The paper introduces an approach for serving and executing RL rollout in multi-turn agent workflows. The premise of the paper is that the state-of-the-art systems deploying multi-agent workload optimize each agent application in isolation. This paper aims to consider the totality of these applications all together and come off with a scheduling solution that will maximize the throughput by selecting which LLMs to evict during the computation, according to the objective of minimizing the cost of re-computation. The program abstraction keeps track of metadata such as context length, execution phase, and tool dependencies. Using this abstraction, the system coordinates GPU KV-cache management with external tool resources. The experimental results suggest throughput gains over vLLM, Continuum, and rollout baselines, along with reduced disk usage.

**Compliance With Llm Reviewing Policy:**

Affirmed.

**Final Justification:**

I am happy with the clarifications that the rebuttal did and I am increasing my score.

**Key Questions For Authors:**

1. What exact assumptions are required for the shortest-first eviction claim to be optimal? As written in the weaknesses, Eq. (7) does not appear to be solved optimally by greedy shortest-first in general. If the result depends on additional assumptions, please state them explicitly.
2. How is the time-decay function f(t) used in the implementation? In Eq. (5), decreasing f(t) reduces the contribution of long-running acting programs to the capacity check, which appears opposite to the textual claim that the system becomes more willing to evict them over time. Please reconcile the equation, the intuition, and the code.
3. Are there any hidden tradeoffs for this higher throughput? Specifically, does it negatively impact success rates, rollout rewards, or the fairness shown to longer-context programs?
4. Can the authors provide more detail on baseline tuning, especially for Continuum and the rollout baseline, to make the comparison easier to evaluate?
5. Can the authors add ablations for global queueing and eviction policy?

**Limitations:**

No. The paper should discuss practical limitations more directly, including fairness for long programs, the dual-use implications of making large-scale agent execution cheaper, etc.

**Strengths And Weaknesses:**

Strengths:
1. The paper tackles a systems bottleneck. As agentic workflows and RL rollouts become more expensive, sustained throughput and resource management become relevant.
2. The proposed system is coherent at a high level. The program abstraction exposes workflow state to the scheduler, and the combination of GPU-memory decisions with tool-lifecycle management is practically meaningful.

Weaknesses:
1. The claim that shortest-first eviction is optimal for Eq. (7) appears to be false as written, as it does not hold in most cases. E.g., if the active programs have context lengths of {1, 100} and the scheduler needs to free ΔC = 100, greedy shortest-first would evict both programs, but evicting only the 100-token program has lower objective value. The optimization problem is misstated and the proof relies on additional assumptions, or the algorithm should be presented as a heuristic rather than an optimum. Appendix E.2 does not resolve the core issue. The proof relies on decomposing a long program into a shorter piece plus a residue, but program eviction is indivisible in the stated problem.
2. The use of the decay term in Eq. (5) contradicts the text's explanation. If f(t) < 1 decreases over time, then long-running acting programs contribute less to the capacity check as time increases, making them theoretically less likely to be evicted. Yet, the text claims the opposite, which says long-idle acting programs become more likely to be evicted over time. This is a central design point, and the math, intuition, and implementation are not well-aligned. Appendix E.3 does not really prove that exponential decay is optimal for throughput. It shows that exponential or geometric decay is the only solution under a specific multiplicative time-homogeneity assumption. That is a modeling assumption, not an optimization result.
3. The evaluation shows that the complete system works well, but it doesn't clearly isolate the main drivers of that success. The paper needs ablations that break down the impact of major ideas like global vs. local queueing, or shortest-first vs. alternative eviction policies.
4. The impact is somewhat specialized. The broader impact depends on whether the system generalizes beyond the specific stack assumed in the paper.
5. Some components feel like adaptations of existing systems ideas rather than entirely new techniques. The authors should draw a sharper line between their novel contributions and the prior work it cites.
6. Appendix F is confusingly written. Appendix E is more assertive than rigorous, and fails to address errors of Eq. (7) and Eq. (5), as stated in Weakness 1 and Weakness 2.

---

> ### Author Rebuttal · Authors · 2026-03-29
>
> We thank the reviewer for thoughtful questions, which help improve the quality of our work.
> # W1, Q1: shortest-first eviction.
> Our scheduling operates on token-level KV cache management, as in vLLM/SGLang/ThunderAgent, so agent-program KV states are divisible at token granularity. Under this setting, freeing 100 tokens means fully evicting the 1-token program and partially evicting 99 tokens from the 100-token program. Pausing a program means holding its future requests to the inference engine but not deleting the KV caches. With this token-level divisibility assumption, shortest-first minimizes the re-prefill STP cost, as shown in Appendix E.2. We will make this assumption explicit in the revision.
>
> # W2, Q2: Mechanism of decay function.
> The role of f(t) is to relax the conservative no-thrashing capacity check, allowing controlled mild thrashing to trade off recomputation STP cost against idle caching STP cost.
>
> As an acting program stays idle longer, its effective memory weight is reduced. This also affects program restoring: under the same capacity budget, lowering the weight of acting programs leaves room to restore more reasoning programs. The decay function allows mild thrashing to balance the tradeoff between recomputation cost and idle caching cost. This tradeoff is also reflected in Figs. 4 and 5, where ThunderAgent can improve throughput even when KV-cache hit rate decreases, by reducing idle caching cost at the expense of limited recomputation. We will revise the text to make this mechanism explicit.
>
> Appendix E.3 is intended to justify exponential decay under unpredictable tool latency, rather than a blanket claim of empirical throughput optimality.
>
> # Q3: Hidden tradeoffs beyond throughput [R2T1](https://anonymous.4open.science/r/anonymous-icml26-submission-B4EF/R2/R2T1.md)
>
> ThunderAgent only changes scheduling and resource management, not model outputs. From our RL experiments in R2T1-1, ThunderAgent improves rollout and reduces task latency, leading to better training quality by improving the success rate. This is because ThunderAgent allows faster rollout and reduces tool costs which makes the agentic RL system more robust.
>
> However, we do observe throughput degradation on single request tasks like GSM8K, where the additional communication buffer and decay function lead to about 10% worse than vLLM's, as shown in R2T1-2.
>
> # W3 and Q5 Ablation study
> Unless otherwise specified, we use the same setup as Table 2 in the paper. Exact settings are given in each figure/table caption.
>
> ## Global and local waiting queue.[Figure1](https://anonymous.4open.science/r/anonymous-icml26-submission-B4EF/R2/R2T2.png)
> We compare our global waiting queue, which allows paused programs to be restored on different GPU nodes, against a local queue that always restores a program to its original node. The global design consistently achieves higher throughput by reducing cross-node memory imbalance.
>
>  ## Decay function.[Figure2](https://anonymous.4open.science/r/anonymous-icml26-submission-B4EF/R2/R2T3.png)
> We compare three representative choices: constant decay (no decay, f(t)=1), linear decay (f(t)=max⁡(1−t,0)), and exponential decay (f(t)=e−t).
>
> We evaluate these three decay rules on three agent pipelines. Mini-SWE-Agent has relatively short and predictable tool execution times. OSWorld introduces longer but still fairly regular waiting times. In contrast, ScienceAgent's execution latency is much less predictable.
>
> Exponential decay outperforms constant and linear decay on Mini-SWE-Agent and ScienceAgent, while trailing linear decay on OSWorld by about 6%. Thus, exponential decay is not uniformly best on every workload, but is the most robust overall across the three pipelines.
>
> ## Ablation of the eviction policy. [Figure3](https://anonymous.4open.science/r/anonymous-icml26-submission-B4EF/R2/R2T4.png)
>
> Shortest-first eviction achieves the highest throughput and the lowest average prefill time, while longest-first eviction performs the worst.
>
> ## Component ablations.
>
> ### R2T2
> |Component|Throughput|
> |:-:|:-:|
> |vLLM|375|
> |local scheduling|602|
> |local scheduling+global queue|672|
>
> The results on two H100 nodes show that the largest throughput gain comes from local scheduling, which directly reduces unnecessary recomputation and idle caching cost. Adding the global waiting queue brings further improvement by reducing memory imbalance across nodes.
>
> # For W4 and W5, please refer to Reviewer1(Ui8N)
>
> # Q4 Experiment Setting
> We would like to clarify that we use the source code and the exact setting reported in Continuum paper for every ablation study. The longest pinning time is set to 2 seconds following the default setting in the [Continuum repository](https://github.com/Hanchenli/vllm-continuum). For the baseline in the distributed rollout experiment, we use SGLang Gateway's [cache-aware](https://docs.sglang.io/advanced_features/sgl_model_gateway.html#cache-aware-policy-tuning) routing policy with default parameters.

---

> > ### Author Rebuttal · Reviewer_bPw6 · 2026-04-02
> >
> > Thank you for your detailed and thoughtful rebuttal. I am happy to change my negative score to positive.

---

> > > ### Author Response · Authors · 2026-04-02
> > >
> > > Thank you very much for your time, effort, and thoughtful feedback throughout the review process! Your careful questions and detailed comments helped us better clarify the paper and improve the overall project. We sincerely appreciate your support.

---

### Official Review · Reviewer_Ui8N · 2026-03-13

**Soundness:** 4
**Presentation:** 4
**Significance:** 4
**Originality:** 3
**Overall Recommendation:** 5
**Confidence:** 3

**Summary:**

This paper proposes ThunderAgent, a program-aware inference system for agentic workflows built on top of existing LLM inference engines and tool orchestration frameworks. The authors investigate an important concept of improving throughput and resource efficiency in multi-turn agentic inference workloads. The system abstracts agent workflows as LLM programs, enabling a program-aware scheduler and tool resource manager that jointly manage KV-cache usage, GPU memory allocation, and external tool environments. Through experiments on multiple agent benchmarks and large models, the paper shows significant throughput improvements over existing systems.

**Compliance With Llm Reviewing Policy:**

Affirmed.

**Final Justification:**

The paper presents a well-motivated and empirically strong system with clear practical significance, and while the originality is moderate, the overall soundness, evaluation quality, and clarity support acceptance. The rebuttal thoroughly addressed my main concerns regarding the program abstraction, scheduling rationale, and robustness, reinforcing my initial positive assessment and final recommendation to accept.

**Key Questions For Authors:**

1. How sensitive is the proposed scheduling mechanism to workload characteristics such as tool execution variability, prompt length growth, or agent reasoning depth? Would performance degrade under significantly different workloads?

2. The system introduces a program abstraction to coordinate scheduling and resource management. Could similar improvements be achieved with modifications to existing inference engines without introducing a new abstraction layer?

3. The paper demonstrates strong throughput improvements, but how does the system affect other important metrics such as latency variance, system stability, or operational complexity in real-world deployments?

4. To what extent do the improvements depend on specific agent frameworks or benchmarks used in the evaluation? Would similar gains be expected in other agentic workloads?

**Limitations:**

No. The paper could discuss practical limitations, such as potential overhead introduced by the program abstraction layer, deployment complexity in heterogeneous production environments, sensitivity to different agent workload characteristics, and scenarios where the scheduling policy may be less effective.

**Strengths And Weaknesses:**

**Strengths**

(+) The paper addresses a timely and practically important problem in LLM infrastructure, namely improving the efficiency of large-scale agentic workflows that involve repeated reasoning steps and tool calls.

(+) The system design is clearly motivated by real bottlenecks in current inference systems, including KV-cache thrashing, memory imbalance across nodes, and inefficient management of tool execution environments.

(+) The experimental evaluation is extensive, covering multiple agent benchmarks, workloads, and hardware configurations, and demonstrates significant throughput improvements over strong baselines.

(+) The paper is generally well written and easy to follow, with clear system diagrams and explanations of the scheduling mechanisms.

(+) If the reported improvements generalize to real deployments, the proposed system could have substantial practical impact on large-scale agentic systems and RL rollout pipelines.


---


**Weaknesses**

(-) While the system achieves strong empirical improvements, the conceptual novelty appears limited, as many components build upon existing inference engines and orchestration frameworks with additional scheduling and resource management layers.

(-) Some design choices and system heuristics appear intuitive but may benefit from deeper analysis or justification regarding why they outperform alternative scheduling strategies.

(-) The evaluation primarily focuses on throughput improvements, while other system metrics such as robustness, scalability across larger clusters, or sensitivity to workload characteristics could be further explored.

---

> ### Author Rebuttal · Authors · 2026-03-30
>
> # W1, Q2 and Reviewer bPw6's W5: Novelty of the program abstraction and the need for a program level design
>
> We agree that ThunderAgent builds on existing inference backends. Our novelty lies not in reimplementing the backend, but in introducing a program-level control plane for agentic inference.
>
> Agent inference spans multiple model calls, tool invocations, and state across GPUs and tool environments; once the application becomes sufficiently complex, this makes a program abstraction above the inference engine necessary for unified scheduling. In ThunderAgent, this layer supports diverse agent workflows without workflow-specific tuning, enables cross-backend load balancing, and coordinates GPU scheduling with tool resource management.
>
> This is also where ThunderAgent differs from prior agent inference systems. Continuum relies on TTL based KV cache pinning rather than program level scheduling and struggles with unpredictable tool latency, while Autellix is program aware but still focuses on GPU scheduling of LLM calls. ThunderAgent instead uses a fuller program abstraction that tracks the full ReAct workflow and its resources, enabling unified scheduling, DP load balancing, and coordinated GPU and tool resource management.
>
> # W2: Cost based motivation of the scheduling policy and justification of the key design choices
>
> We agree that the motivation behind these design choices can be made more explicit.
>
> Our scheduling policy is not intended as a collection of intuitive heuristics. Instead, it is derived from the cost decomposition in Section 4.2, where the scheduling objective is to minimize recomputation, idle caching, and unused capacity. Periodic monitoring is introduced because thrashing can emerge dynamically as contexts grow during execution, not only when a new request arrives.
>
> The shortest first eviction rule is also theory-driven. Appendix E.1 shows that recomputation STP cost grows quadratically with context length, and Appendix E.2 then proves, via an exchange argument, that evicting the shortest programs minimizes the total recomputation penalty under a fixed memory reclamation target. The time decay term addresses a different tradeoff: whether to keep the cache of acting programs or to reclaim that memory for useful decoding. Appendix E.3 further shows that, under our assumptions, exponential decay is the only admissible form of f(t).
>
> More broadly, these design choices are guided by the same cost-based analysis, rather than introduced as separate heuristics.
>
> # W3 and Q3: Scalability, workload robustness, system stability, and low operational complexity
>
> We agree that robustness under diverse workloads, scalability, system stability, and deployment complexity are all important dimensions for evaluating a practical agentic inference system, in addition to throughput.
>
> First, regarding scalability, we have added 2–8 node rollout results in Table [R4T4](https://anonymous.4open.science/r/anonymous-icml26-submission-B4EF/R4/R4T4.md) of our response to Reviewer RukN, showing that ThunderAgent scales beyond the main paper’s 2-node setting.
>
> Second, regarding sensitivity to workload characteristics, our evaluation already spans substantially different agent workflows, including coding agents, remote API based workloads, and scientific discovery agents with more stochastic tool behavior. ThunderAgent consistently improves throughput across them, as shown in Figure 4.
>
> Third, regarding system stability, ThunderAgent explicitly manages the lifecycle of tool environments. Hook based garbage collection prevents resource leakage over long running workloads, while asynchronous environment preparation reduces blocking overheads during execution; these effects are illustrated in Figures 2b and 2c.
>
> Finally, regarding operational complexity, ThunderAgent is a lightweight middle layer between the inference engine and agent workflows, and in our implementation it requires only two lines of code to integrate, as described in Appendix B.3.
>
> # Q1, Q4 and Reviewer bPw6's W4: Adaptation to workload variation and generalization beyond the evaluated frameworks
>
> ThunderAgent is primarily designed for ReAct-style agentic workflows, where the model repeatedly alternates between reasoning and acting over a multi-turn trajectory. This setting is widely used in modern agent systems, including Codex, OpenClaw, and DeepSearch, and covers a broad range of agent applications. Our evaluation reflects this breadth, spanning coding, tool orchestration, and scientific discovery.
>
> Within this setting, ThunderAgent is designed to adapt to workload variation rather than rely on fixed workload assumptions. Through the program abstraction, the scheduler reasons over program state, so its decisions do not depend on preset tool durations or trajectory growth patterns. This is consistent with our results, which show consistent throughput gains across workloads with substantially different tools and execution patterns.

---

> > ### Author Rebuttal · Reviewer_Ui8N · 2026-04-03
> >
> > Thank you for the clear and detailed rebuttal. The response adequately addresses my main questions about the program abstraction, the cost-based scheduling rationale, and the system’s robustness and scalability, with helpful clarifications and additional evidence. Overall, my original concerns are sufficiently addressed. I maintain my score of recommending the acceptance.

---

### Decision · Program_Chairs · 2026-04-30

**Decision:**

Accept (spotlight)

**Comment:**

This paper introduces ThunderAgent, a program-aware inference system designed to optimize the execution of multi-turn agentic workflows. By abstracting agent workflows as "LLM Programs," the system coordinates GPU KV-cache management and external tool resources to minimize the costs associated with recomputation and idle caching. Using a program-aware scheduler and a global waiting queue, it dynamically balances memory across nodes and prepares tool environments asynchronously, aiming to significantly improve throughput and reduce disk usage during periods when agents are waiting for external tools to execute.

Reviewers unanimously recognized that the paper addresses a timely and practically critical bottleneck in large-scale agentic infrastructure and RL rollouts (Ui8N, bPw6). The system design was praised for directly tackling real-world inefficiencies like KV-cache thrashing and cross-node memory imbalance using a sensible program-level abstraction (Ui8N, RukN). Furthermore, the reviewers commended the extensive experimental evaluation, which demonstrated strong empirical throughput improvements across diverse, realistic agent benchmarks, successfully handling the stochastic nature of external tool execution (Ui8N, atjC, RukN).

Initial weaknesses raised by the reviewers included questions about the conceptual novelty of building on existing engines (Ui8N, bPw6), the mathematical rigor behind the shortest-first eviction policy and time-decay functions (bPw6), and the system's robustness to unpredictable tool latency and hardware scalability (atjC, RukN). In the rebuttal, the authors successfully clarified the theoretical foundations, explaining that token-level divisibility makes the shortest-first strategy optimal for minimizing re-prefill costs and that exponential decay acts as a principled counter to unpredictable latencies (bPw6, atjC). They also provided compelling new data showing the system scales effectively up to 8 nodes and remains highly effective even when combined with KV-cache offloading on hardware with lower compute-to-bandwidth ratios (RukN).

Overall, I recommend this paper for acceptance.